# Turnip mosaic virus co-opts the vacuolar sorting receptor VSR4 to promote viral genome replication in plants by targeting viral replication vesicles to the endosome

Guanwei Wu[1], Zhaoxing Jia[1,2], Kaida Ding[1], Hongying Zheng[1], Yuwen Lu[1], Lin Lin[1], Jiejun Peng[1], Shaofei Rao[1], Aiming Wang[3], Jianping Chen[1]*, Fei Yan[1]*

**1** State Key Laboratory for Managing Biotic and Chemical Threats to the Quality and Safety of Agro-products, Institute of Plant Virology, Ningbo University, Ningbo, China, **2** College of Plant Protection, Nanjing Agricultural University, Nanjing, China, **3** London Research and Development Centre, Agriculture and Agri-Food Canada, London, Ontario, Canada

\* jianpingchen@nbu.edu.cn (JC); yanfei@nbu.edu.cn (FY)

## Abstract

Accumulated experimental evidence has shown that viruses recruit the host intracellular machinery to establish infection. It has recently been shown that the potyvirus *Turnip mosaic virus* (TuMV) transits through the late endosome (LE) for viral genome replication, but it is still largely unknown how the viral replication vesicles labelled by the TuMV membrane protein 6K2 target LE. To further understand the underlying mechanism, we studied the involvement of the vacuolar sorting receptor (VSR) family proteins from Arabidopsis in this process. We now report the identification of VSR4 as a new host factor required for TuMV infection. VSR4 interacted specifically with TuMV 6K2 and was required for targeting of 6K2 to enlarged LE. Following overexpression of VSR4 or its recycling-defective mutant that accumulates in the early endosome (EE), 6K2 did not employ the conventional VSR-mediated EE to LE pathway, but targeted enlarged LE directly from cis-Golgi and viral replication was enhanced. In addition, VSR4 can be *N*-glycosylated and this is required for its stability and for monitoring 6K2 trafficking to enlarged LE. A non-glycosylated VSR4 mutant enhanced the dissociation of 6K2 from cis-Golgi, leading to the formation of punctate bodies that targeted enlarged LE and to more robust viral replication than with glycosylated VSR4. Finally, TuMV hijacks *N*-glycosylated VSR4 and protects VSR4 from degradation via the autophagy pathway to assist infection. Taken together, our results have identified a host factor VSR4 required for viral replication vesicles to target endosomes for optimal viral infection and shed new light on the role of *N*-glycosylation of a host factor in regulating viral infection.

## Author summary

A key feature of the replication of positive-strand RNA viruses is the rearrangement of the host endomembrane system to produce a membranous replication organelle. Recent

**Data Availability Statement:** All relevant data are within the manuscript and its Supporting Information files.

**Funding:** This work was supported by grants from the National Natural Science Foundation of China (32070165) to G.W, Chinese Agriculture Research System (CARS-24-C-04) to F.Y., the Ningbo Major Special Projects of the Plan "Science and Technology Innovation 2025" (2021Z106) to H.Z, and sponsored by the K.C. Wong Magna Fund in Ningbo University. The funders had no role in study design, data collection and analysis, decision to publish, or preparation of the manuscript.

**Competing interests:** The authors have declared that no competing interests exist.

reports suggest that the late endosome (LE) serves as a replication site for the potyvirus *Turnip mosaic virus* (TuMV), but the mechanism(s) by which TuMV replication vesicles target LE are far from being fully elucidated. Identification of the host factors involved in this transport process could lead to new strategies to combat TuMV infection. In this report, we provide evidence that TuMV replication depends on functional vesicle transport from cis-Golgi to the enlarged LE pathway that is mediated by a specific VSR family member, VSR4, from Arabidopsis. Knock out of *VSR4* impaired the targeting of TuMV replication vesicles to enlarged LE and suppressed viral infection, and this process depends on the specific interaction between VSR4 and the viral replication vesicle-forming protein 6K2. We also showed that *N*-glycosylation of VSR4 modulates the targeting of TuMV replication vesicles to enlarged LE and enhances viral infection, thus contributing to our understanding of how TuMV manipulates host factors in order to establish optimal infection. These results may have implications for the role of VSR in other positive-strand RNA viruses.

## Introduction

Eukaryotic cells are compartmentalized by various membrane-bound organelles that form a complex endomembrane network to perform diverse fundamental functions critical for cell survival. This dynamic endomembrane system is maintained through the continuous flux of vesicles to exchange lipids and proteins. Plant cells all have an endomembrane system, including plasma membrane (PM), nuclear envelope, the endoplasmic reticulum (ER), cis-Golgi apparatus, *trans*-Golgi network or early endosome (TGN/EE), prevacuolar compartment/multi-vesicular body or late endosome (PVC/MVB/LE) and vacuole. The secretory and endocytic pathways are two major transport routes of the plant endomembrane system [1]. These two pathways merge in the TGN/EE and their cargoes are passed on to the PVC/MVB/LE by different sorting machineries. PVC/MVB/LE serve as intermediate compartments between the TGN/EE and the vacuole, and enable proteins to recycle before fusion with the vacuole [2]. Plant viruses are obligate parasites with genomes that encode very few proteins and they exploit the host intracellular machinery to establish infection [3–5]. In the past decade, an essential role for the endomembrane system (including related host factors) in plant virus infection has been recognised [6–20]. In recent reports, PVC/MVB/LE were observed to be labelled with double stranded RNA (dsRNA) during infection by *Turnip mosaic virus* (TuMV; genus *Potyvirus*) [9,10], suggesting that TuMV replication vesicles transit through PVC/MVB/LE. The endosomal sorting complexes required for transport (ESCRT) family proteins, normally required for PVC/MVB/LE formation, are also hijacked by *Brome mosaic virus* (BMV, genus *Bromovirus*) and *Tomato bushy stunt virus* (TBSV, genus *Tombusvirus*) to facilitate VRC assembly [5,17,21–23], indicating an important role of PVC/MVB/LE during plant virus infection.

The genus *Potyvirus* is the largest genus of known plant RNA viruses, comprising more than 180 species, constituting about 15% of all identified plant viruses [24,25]. Potyviruses include some of the most economically important plant pathogens, including TuMV, *Soybean mosaic virus*, *Potato virus Y*, and *Plum pox virus* [26]. Potyviruses have a single-stranded positive-sense (+ss) RNA genome of approximately 10, 000 nucleotides that encodes a long open reading frame (ORF) and a small ORF P3N-PIPO resulting from RNA polymerase slippage [24,27]. The large ORF is proteolytically processed by three viral proteinases into 10 mature proteins, one of which is the membrane-associated 6K2 protein. 6K2 can remodel the host ER

for the formation of viral replication vesicles [28,29]. These vesicles contain viral RNA and viral proteins as well as host components [18,29]. These vesicles may mature into replication-competent single membrane vesicles, which can fuse with chloroplasts for efficient replication [28,30]. These vesicles are believed to take a Golgi by-pass unconventional pathway and reach PVC/MVB/LE for efficient virus infection [10]. It has not been established how 6K2 vesicles reach PVC/MVB/LE but they cannot utilize the post-Golgi trafficking pathway [8,31].

Vacuolar sorting receptors (VSRs) are arguably the most studied vacuolar trafficking factors in plants, but it is still unclear how they function in the plant endomembrane system [32–36]. The genome of *Arabidopsis thaliana* contains seven VSR isoforms (AtVSR1-7) with highly conserved amino acid sequences especially in their C-terminal domains [37]. All seven AtVSRs localize to PVC/MVB/LE [38]. AtVSRs are expressed in most Arabidopsis tissue types, including root, leaf, stem, flower, pollen, and seed [39], but the different AtVSRs are not equally expressed in each of these tissues, suggesting that they might have distinct functions. For example, AtVSR1 and 4 are predominantly expressed in leaves, while AtVSR5, 6, and 7 are only expressed in roots, and AtVSR3 is expressed specifically in guard cells [33,39]. VSRs have an N-terminus luminal domain, a single transmembrane domain (TMD) and a C-terminus cytoplasmic tail (CT). The CT carries the sorting information for their own transportation and harbours a YXXΦ motif ($^{610}$YMPL$^{613}$) and an acidic dileucine-like motif ($^{602}$EIRAIM$^{607}$). The YXXΦ motif was interpreted as an anterograde TGN to PVC signal, and an AMPA mutant was shown to accumulate in TGN [40,41]. In contrast, the acidic dileucine-like motif was suggested to act as a retrograde sorting motif from PVC to TGN, but also as an endocytosis signal, and AIRAAA mutants led to strong labelling of the vacuole [41,42]. According to a frequently cited model, the cargo-VSR complex is sequestered into clathrin coated vesicles at the TGN and delivered to the PVC. However, evidence is accumulating showing that VSR-cargo interactions occur much earlier in the endomembrane system. According to textbooks and most recent reviews, VSRs recognize their ligands in the ER [43]. The receptor-ligand complexes are then transported in COPII-coated vesicles to Golgi stacks, and eventually reach the TGN/EE via cisternal progression [34]. The ligands then dissociate from the VSRs, which are recycled back to cis-Golgi cisternae mediated by the retromer complex [34,44,45]. Interestingly, the VSR CT associates with a PVC/MVB/LE SNARE protein VTI11 (VESICLE TRANSPORT V-SNARE11) [46], which also copurifies with TuMV 6K2 and is required for 6K2 vesicle trafficking to PVC/MVB/LE [10]. Thus, we questioned whether VSR participates in 6K2 targeting of PVC/MVB/LE during TuMV infection.

In this study, we demonstrate that the *Arabidopsis thaliana* VSR4 (AtVSR4) interacts with TuMV 6K2 specifically and is an essential proviral host factor for TuMV infection. Moreover, we reveal that AtVSR4 can be *asparagine* (*N*)-glycosylated and monitor 6K2 traffic to PVC/MVB/LE. We further explore the role played by AtVSR4 and its *N*-glycosylation in TuMV infection.

## Results

### AtVSR4 is required for TuMV infection

To verify whether VSR proteins are involved in TuMV infection, we obtained homozygous knockout (*ko*) transfer DNA lines for each of five AtVSR homologs (*AtVSR1*, *AtVSR2*, *AtVSR4*, *AtVSR5* and *AtVSR7*) and used them in viral infection assays (S1 Fig). All these *atvsr* mutants had similar growth and developmental phenotypes to the wild-type Col-0 plants. Homozygous *atvsr ko* and Col-0 plants were mechanically inoculated with TuMV-GFP and then monitored for green fluorescence using a hand-held UV lamp. TuMV-GFP fluorescence was first seen in the inoculated Col-0 and mutants' leaves at 4 days post inoculation (dpi). We

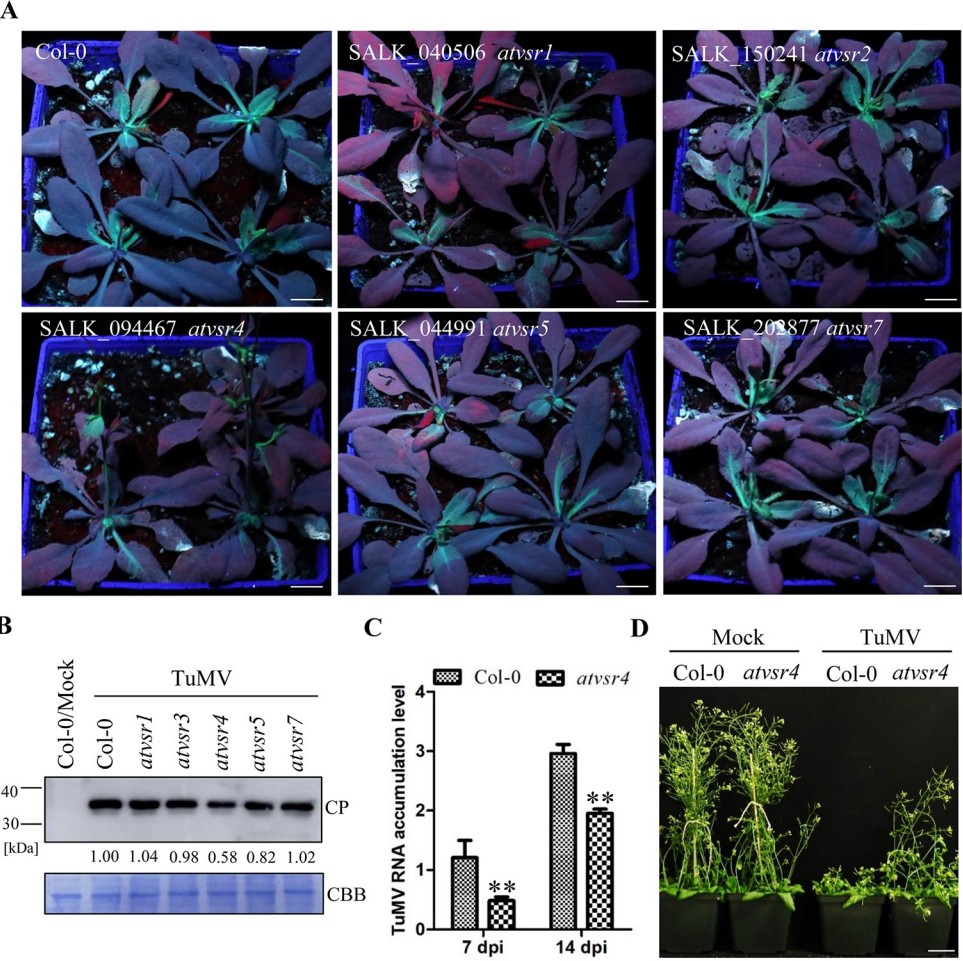

**Fig 1. AtVSR4 is required for TuMV infection.** (A) Representative images of TuMV-GFP infected wild-type Col-0 and *atvsr* mutant plants at 7 dpi under hand-held UV lamp. Scale bar = 1.5 cm. (B) Immunoblots detecting TuMV CP in the systemic leaves at 7 dpi. The relative TuMV CP signals were quantified by ImageJ software. Coomassie Brilliant Blue (CBB) R-250-stained RuBisco large subunit serves as a loading control. TuMV CP was detected with anti-TuMV CP polyclonal antibody. (C) qRT-PCR analysis of the TuMV RNA levels in the wild-type Col-0 and *atvsr4* mutant plants at 7 and 14 dpi. Actin II was used as an internal control. Error bars represent the standard deviation of three biological replicates of a representative experiment. Statistical analysis was performed using Student's t-test (**, $P < 0.01$). (D) Phenotypes of the wild-type Col-0 and *atvsr4* plants inoculated with TuMV and the mock infection controls at 16 dpi. Scale bar = 5.0 cm.

recorded the GFP fluorescence in all plants at 7 dpi. Plants of the *atvsr4* line had much milder symptoms and weaker GFP fluorescence than the Col-0 and other *atvsr* plants (Fig 1A). Immunoblots revealed substantially reduced levels of TuMV coat protein (CP) at 7 dpi in *atvsr4* than in Col-0 and the other *atvsr* mutants (Fig 1B). At both 7 dpi and 14 dpi, TuMV RNA levels in *atvsr4* were significantly lower that those in Col-0 plants (Fig 1C), and at 16 dpi the symptoms in *atvsr4* remained much milder than those in Col-0 plants (Fig 1D). These combined results indicate that AtVSR4 is required for TuMV infection.

## AtVSR4 interacts specifically with TuMV 6K2

To understand how AtVSR4 participates in TuMV infection, we examined possible protein-protein interactions between AtVSR4 and each of the 11 TuMV-encoded proteins. Since most

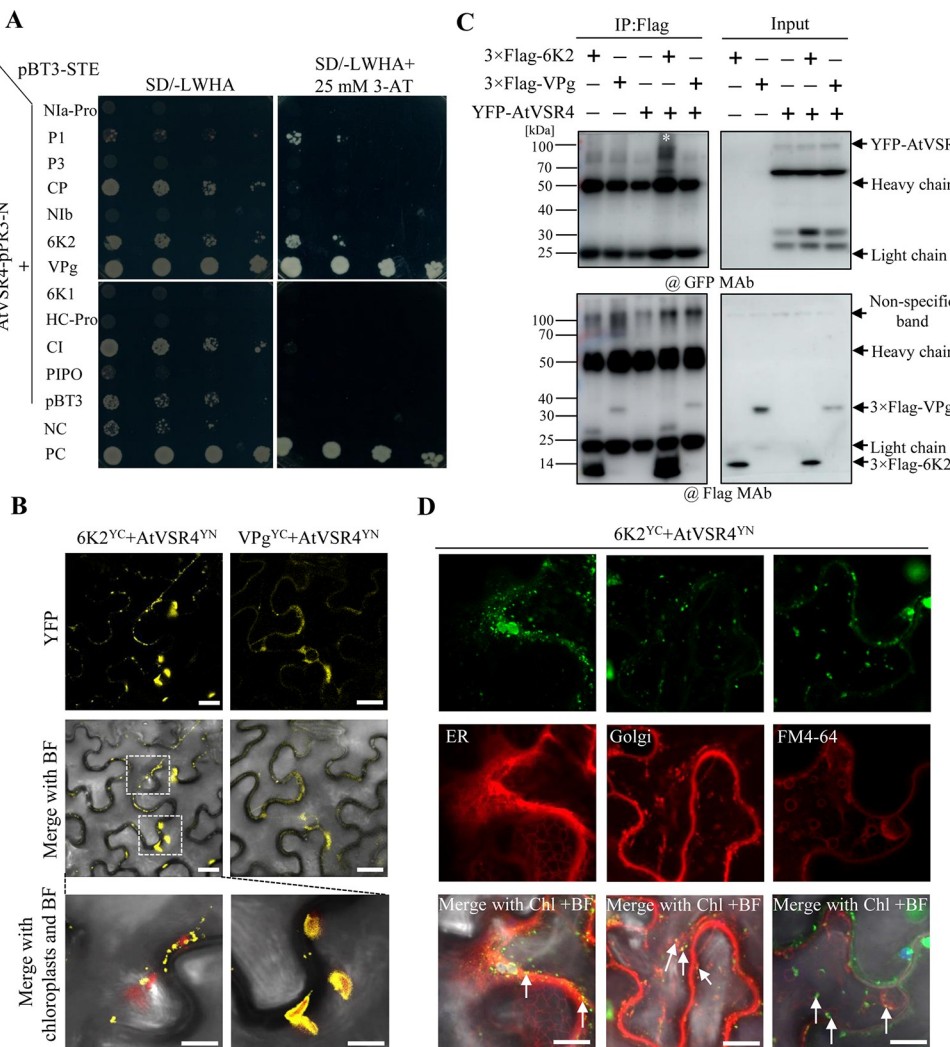

**Fig 2. AtVSR4 interacts with TuMV 6K2.** (A) Protein-protein interaction assay between AtVSR4 and each of 11 TuMV proteins using a membrane yeast two-hybrid (Y2H) method. This experiment was repeated twice and representative pictures are shown. NC, negative control. PC, positive control. (B) Images from a BiFC assay *in planta* to confirm the interactions detected by Y2H. The interactions between AtVSR4 and TuMV 6K2 or VPg were confirmed in *N. benthamiana* cells. Pictures were taken at 2 d post agroinfiltration (dpai). The YFP field and overlay of YFP with BF (bright field) are shown. Scale bar = 20 µm. (C) Results of co-IP assay showing that AtVSR4 can form complexes with TuMV 6K2 in *N. benthamiana* cells. Different cell lysates were immunoprecipitated with anti-Flag M2 gel beads, separated by SDS-PAGE and immunoblotted with anti-Flag monoclonal antibody (@Flag MAb), or anti-GFP monoclonal antibody (@GFP MAb). Black asterisk denotes YFP-fusion AtVSR4 band captured by 6K2. (D) Colocalization of the BiFC signals from TuMV 6K2YC and AtVSR4YN with ER marker (mCherry-HDEL), Golgi marker (Man49-mCherry) and endosomal compartments labelled by FM4-64 in *N. benthamiana* cells. Chl, auto-fluorescent chloroplasts. BF, bright field. Scale bar = 20 µm.

of TuMV proteins and AtVSR4 are membrane-associated or located in cytosol, the nucleus-based yeast two hybrid system is not appropriate for detecting protein-protein interactions. We therefore used a split-ubiquitin membrane-based yeast two hybrid (Y2H) system as previously described [8]. Each of the 11 TuMV genes and *AtVSR4* were cloned into the bait vector pBT3-STE and the prey vector pPR3-N, respectively. All of the NMY51 yeast cells co-expressing the TuMV bait proteins with the prey AtVSR4 or empty prey plasmids grew robustly on the double dropout medium, SD/-Leu/-Trp (Fig 2A). Yeast cells co-transfected with the

TuMV 6K2 bait and AtDRP1A prey served as the positive control [9]. Transformants were then selected on plates lacking leucine, tryptophan, histidine and adenine (SD-LWHA). Background growth due to leaky HIS3 gene expression was suppressed by adding 3-aminotriazole (3AT) to concentrations of 25 mM. The results suggested that AtVSR4 may interact with TuMV 6K2, VPg and P1 (Fig 2A).

We further examined the interactions between AtVSR4 and the TuMV proteins 6K2, VPg and P1, in *Nicotiana benthamiana* cells using bimolecular fluorescence complementation (BiFC) analysis. Confocal laser scanning microscopy suggested that there was no interaction between AtVSR4 and P1 (S2 Fig), but there were strong reconstituted YFP fluorescence signals in the combination TuMV 6K2 and AtVSR4, while VPg and AtVSR4 gave a much weaker signal (Fig 2B). AtVSR4 bound with 6K2 at the outer membrane of chloroplasts, a typical 6K2 subcellular localization [28], as revealed in the magnified view (Fig 2B). In addition, the interaction complex of AtVSR4 with 6K2 showed many discrete granules in the cytoplasm, while AtVSR4 interacted with VPg in the cytoplasm close to the PM. Negative controls did not show any fluorescent signals (S2 Fig).

Finally, to confirm the positive interactions identified, we performed a co-immunoprecipitation (co-IP) experiment by co-expressing 3×flag-tagged 6K2 or VPg and YFP-tagged AtVSR4 in a transient assay in *N. benthamiana* leaves as previously described [9]. The co-IP data failed to confirm the AtVSR4 and VPg interaction (Fig 2C), suggesting that this interaction, if any, might be transient or unstable. However, we validated the interaction between AtVSR4 and 6K2 (Fig 2C). We then further verified the localization of the 6K2-VSR4 interaction complex. Confocal microscopy showed that the discrete granules resulting from the 6K2-VSR4 interaction in the cytoplasm colocalized with the single membrane vesicle-like structures (SMVLs) highlighted by N-(3-triethylammoniumpropyl)-4-(6-(4-(diethylamino) phenyl) hexatrienyl) pyridinium dibromide (FM4-64) (Fig 2D), a styryl dye used to monitor PM and endosome localization [47]. These SMVLs labelled by FM4-64 have been shown to be PVC/MVB/LE [2,48]. Moreover, the granules clearly localized at both endoplasm reticulum (ER) and cis-Golgi, as revealed by the colocalization signals with the ER marker mCherry-HDEL and cis-Golgi marker Man49-mCherry [49] (Fig 2D). The results therefore demonstrate that AtVSR4 interacts with TuMV 6K2 specially at several compartments, including chloroplasts, ER, cis-Golgi, and enlarged LE. Since 6K2 alone can localize to chloroplasts, it is probably expected that the VSR4-6K2 complex would also target chloroplasts. In this study, we aimed to investigate how 6K2 targets LE, and whether VSR participates in this progress. We therefore focused on the biological significance of the 6K2-VSR4 complex in LE in the subsequent experiments.

To map the domains of AtVSR4 that bind to TuMV 6K2, we constructed truncated mutants of AtVSR4 for use in a BiFC assay. AtVSR4 has an N-terminus luminal domain (LD), a single transmembrane domain (TMD) and a C-terminus cytoplasmic tail (CT). Thus, we first divided AtVSR4 into three corresponding fragments (S3A Fig). Only CT interacted with 6K2 (S3B Fig) and we therefore then constructed a series of truncated mutants in CT, and found that the [597]QYMDS[601] motif in AtVSR4 CT was responsible for binding to 6K2 (S3B Fig). No positive interaction was found for other combinations or in the negative controls. To further confirm the direct role of AtVSR4-6K2 interaction in TuMV infection, a VSR4 mutant with the binding motif [597]QYMDS[601] mutated to [597]AAAAA[601], namely AtVSR4-C1A (S3A Fig), was generated. A co-IP assay confirmed that AtVSR4-C1A did not bind with 6K2 (S3C Fig). Transient overexpression of AtVSR4-C1A resulted in viral accumulation similar to that in the GUS control, whereas AtVSR4 substantially promoted viral infection (S3D–S3F Fig). These results demonstrate that the AtVSR4-6K2 interaction is required for TuMV infection.

## AtVSR4 colocalizes with the viral 6K2-induced viral replication complex (VRC) and is required for the targeting of 6K2 to enlarged LE

We then questioned whether VSR4 colocalized with 6K2 in planta. Irrespective of the particular fluorescent protein fused to its N or C terminus, AtVSR4 formed many granules close to the PM (S4A Fig). Many of these granules were tightly associated with, or inside, the endosomes highlighted by FM4-64 (Fig 3A), and rarely colocalized with cis-Golgi (S4B Fig, upper panels). Moreover, AtVSR4 also formed ER-like reticular pattern structures (S4A Fig) and merged well with ER marker mCherry-HDEL (S4B Fig, lower panels). We further examined the colocalizations of AtVSR4 and 6K2 in plant cells. AtVSR4-labelled punctate bodies colocalized with the granules formed by 6K2, and these colocalization signals were tightly associated with or within FM4-64 labelled enlarged LE (Fig 3B). Thus, the distribution of AtVSR4 was dynamic and the AtVSR4-6K2 complex colocalized at enlarged LE.

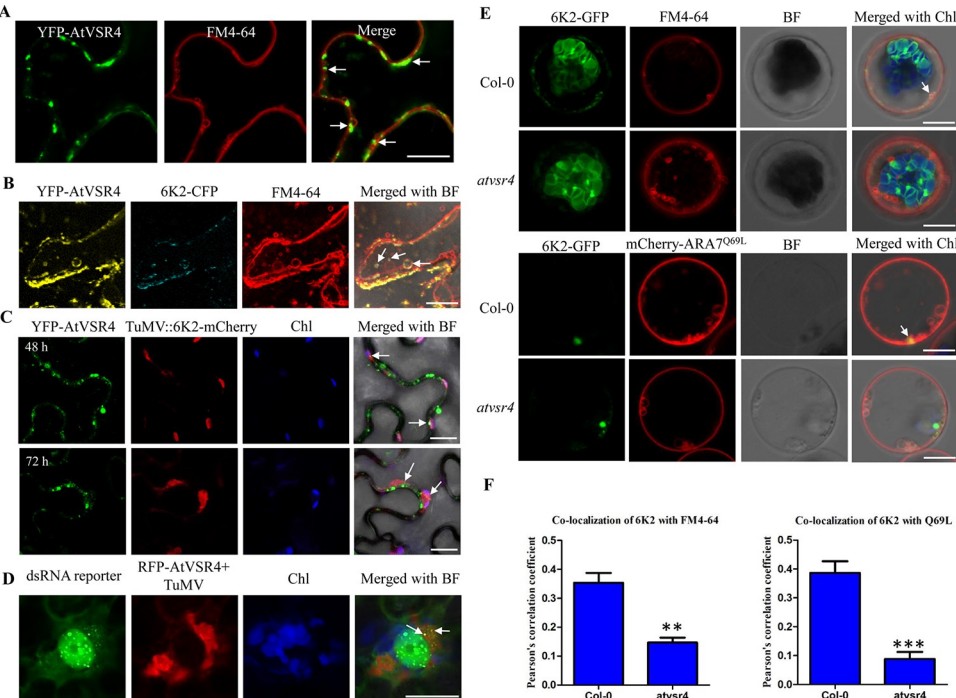

**Fig 3. AtVSR4 is recruited into the viral replication complex induced by TuMV 6K2 and required for enlarged PVC targeting of 6K2.** (A) Colocalization assay of YFP-AtVSR4 with endosomal compartments labelled by FM4-64 in *N. benthamiana* cells at 2 dpai. Scale bar = 20 μm. (B) Colocalization assay of YFP-AtVSR4 with 6K2-CFP and endosomal compartments labelled by FM4-64 in *N. benthamiana* cells at 2 dpai. White arrows indicate that AtVSR4-labelled punctate bodies colocalized with the granules formed by 6K2, and these colocalization signals were tightly associated with, or within, enlarged LE stained by FM4-64. (C) AtVSR4 colocalized with 6K2-induced vesicles in TuMV-infected *N. benthamiana* cells. Localizations of GFP-tagged AtVSR4 in the leaf cells inoculated with TuMV/6K2mCherry at 48 or 72 h post agroinfiltration are shown. (D) AtVSR4 colocalized with TuMV viral replication complex in the dsRNA reporter *N. benthamiana* B2-GFP cells. Images show the localization of GFP-tagged AtVSR4 in leaf cells inoculated with TuMV/6K2gus at 72 h post agroinfiltration. Representative colocalization signals are indicated with white arrows. Scale bar = 20 μm. Chl, auto-fluorescent chloroplasts. BF, bright field. (E) Knockout of *AtVSR4* interferes with the trafficking of TuMV 6K2 to enlarged LE. Colocalization of TuMV 6K2-GFP with FM4-64 or enlarged PVC marker mCherry-AtARA7$^{Q69L}$ in Arabidopsis protoplasts from wild-type Col-0 and *atvsr4* plants are shown. Representative colocalizations are indicated with white arrows. Images were taken at 20 h post transfection. Scale bar represents 10 μm. (F) Quantification of colocalization between 6K2 and FM4-64 or enlarged LE marker in Col-0 or *atvsr4* protoplasts by calculation of the Pearson's correlation coefficient (PCC) values. PCC was measured from 20 protoplasts. Mean values ± standard deviation from three independent experiments are shown. Statistical analysis was performed using Student's t-test ($^{**}$, $P < 0.01$; $^{***}$, $P < 0.001$).

To investigate whether AtVSR4 is recruited by the 6K2-induced viral replication complex (VRCs) for TuMV infection, we transiently expressed AtVSR4-GFP in *N. benthamiana* leaf tissues co-agroinfiltrated with a modified recombinant TuMV infectious clone TuMV::6K2-mCherry, where an extra 6K2-mCherry fusion is inserted between P1 and HC-Pro. It is known that potyvirus replication takes place in the 6K2-induced vesicle structures [28,29]. We found that AtVSR4-induced punctate bodies colocalized with the 6K2-induced membranous structures during TuMV infection at two examined time points (Fig 3C). To confirm that AtVSR4 indeed targeted 6K2-containing VRCs, we further expressed RFP-AtVSR4 transiently in the B2:GFP double stranded RNA (dsRNA) reporter *N. benthamiana* leaf tissues [50], infected with TuMV. DsRNA signals resulting from VRCs containing 6K2 clearly colocalized with RFP-tagged AtVSR4 (Fig 3D). RFP-AtVSR4 did not colocalize with dsRNA signals in the absence of viral infection (S4C Fig). These results suggest that AtVSR4 is recruited into the TuMV VRC.

Finally, to explore whether AtVSR4 is required for enlarged LE targeting of 6K2, we investigated the subcellular distribution of 6K2 in the *atvsr4* mutant. In wild-type Arabidopsis

Col-0 protoplasts, some FM4-64 stained enlarged LE were labelled by TuMV 6K2 but in *atvsr4* protoplasts, almost no enlarged LE colocalized with TuMV 6K2 (Fig 3E). We further used an enlarged LE marker mCherry-ARA7$^{Q69L}$ [51], to colocalize with 6K2 in Col-0 or *atvsr4* protoplasts, which gave similar results. The Pearson's correlation coefficient (PCC) values [52], which provide a quantitative estimate of colocalization, were calculated and confirmed these observations (Fig 3F). These results suggest that AtVSR4 is required for the targeting of TuMV 6K2 to enlarged LE.

## AtVSR4 facilitates TuMV 6K2 to target enlarged LE for viral replication

Recent studies have shown that VRCs of TuMV induced by 6K2 were associated with PVC/MVB/LE and that this required a host factor VTI11 [9,10]. Since VSR can also target endosomes and is associated with VTI11, we hypothesised that 6K2 may utilize the VSR-mediated trafficking pathway. We used a dominant negative inhibition approach to analyze VSR traffic during TuMV infection. As described in the introduction, the VSR4 CT harbors YXXΦ ($^{610}$YMPL$^{613}$) and dileucine-like ($^{602}$EIRAIM$^{607}$) motifs for VSR transportation [33] (Fig 4A). When the YXXΦ motif is mutated to AMPA, the anterograde TGN to PVC signal is disrupted, and this mutant mainly accumulates in TGN [40]. Mutation of the dileucine-like motif to AIRAAA interferes with the PVC to TGN signal, leading to strong labelling of this mutant in the vacuole [42].

We transiently co-expressed *AtVSR4* and the mutants AMPA, AIRAAA, or double mutant (designated as DM) with TuMV::GFP in *N. benthamiana* leaves and monitored the viral infection by measuring both RNA and protein accumulation. Under UV light, TuMV-GFP fluorescence was brighter in the leaf tissues co-infiltrated with TuMV infectious clone and AtVSR4 or its mutants, compared with the control (S5A Fig). A qRT-PCR assay of TuMV genomic RNA [either sense (+) or negative sense (-)] was further performed to evaluate viral replication levels at 60 h post agroinfiltration (hpai). TuMV replicates in the primarily infected cells and viral cell-to-cell movement usually does not occur until 96 hpai [53]. Consistently, significantly higher TuMV (+) or (-) RNA levels were detected in the leaf tissues agroinfiltrated with TuMV together with either VSR4 or its mutants at 60 hpai, compared with that in the control agroinfiltrated with TuMV and partial GUS (ß-glucuronidase)-myc (S5B Fig). Moreover, overexpression of AMPA and DM mutants led to significantly higher viral RNA levels than those in the VSR4 and AIRAAA mutant treatments. Further immunoblotting analysis was performed to evaluate the viral CP accumulation level at 72 hpai. Consistently, much stronger TuMV CP

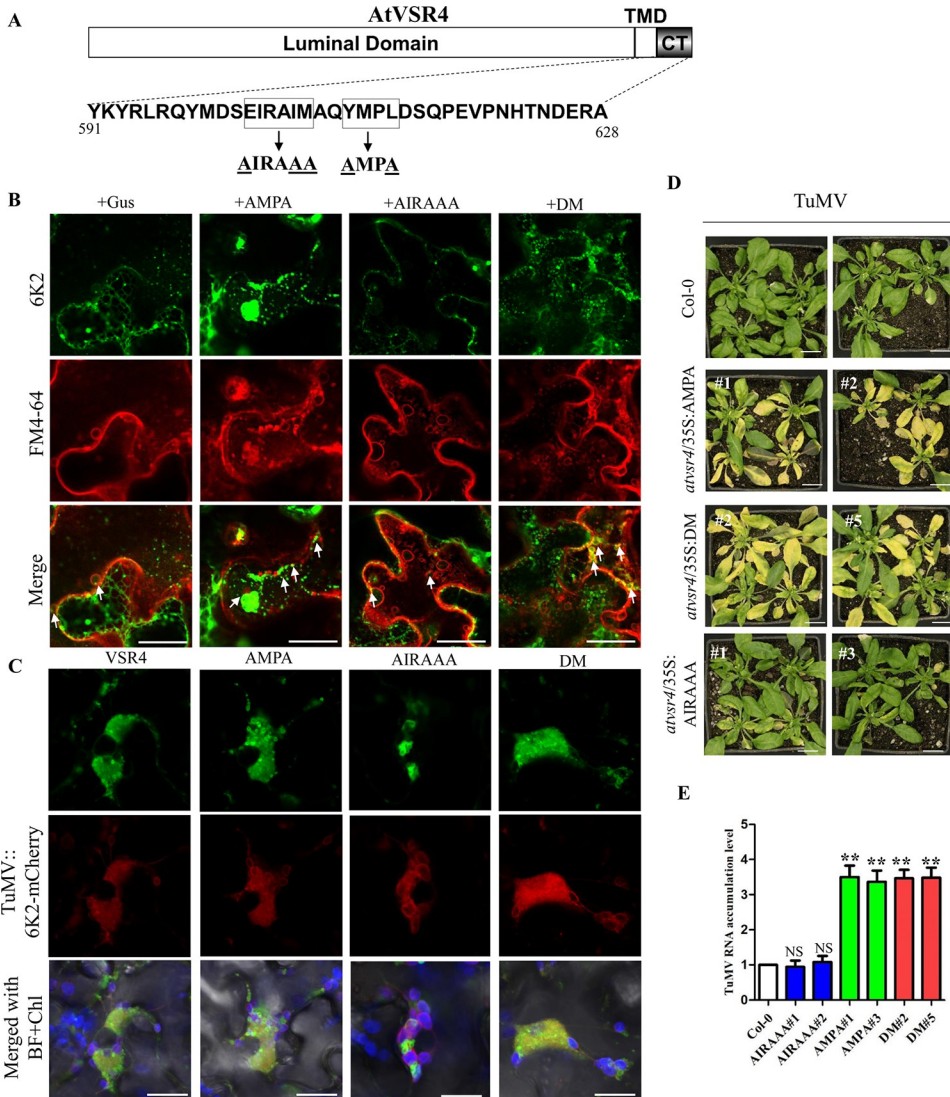

**Fig 4. Effects of mutations in the YXXΦ or acidic dileucine-like motif of AtVSR4 on TuMV infection.** (A) Diagram showing the mutated amino acids of two conserved motifs in the CT region of AtVSR4 used in this study. The complete amino acid sequence of VSR4 CT is shown. The YXXΦ motif YMPL and acidic dileucine-like motif EIRAIM are highlighted with black rectangular boxes, and the mutated amino acids in these motifs are underlined. (B) Effect of wild type AtVSR4 or its mutants on 6K2 localization in *N. benthamiana* cells. YFP-tagged 6K2 was co-expressed with AtVSR4, its mutants or GUS control fused with a c-myc tag. The YFP-6K2 signals were stained with FM4-64. White arrows indicate the enrichment of YFP-6K2 in FM4-64 labelled single membrane vesicle-like structures (SMVLs). Scale bar represents 20 μm. BF, bright field. (C) Colocalization assay of AtVSR4 or its mutants with TuMV 6K2-induced vesicles in *N. benthamiana* cells. Localizations of GFP-tagged AtVSR4 and mutants in the leaf cells inoculated with TuMV/6K2mCherry at 72 h post agroinfiltration are shown. Chl, auto-fluorescent chloroplasts. (D) Phenotypes of the wild-type Col-0, and transgenic plants overexpressing AtVSR4 mutants in the *atvsr4* background, inoculated with TuMV at 14 dpi. Scale bar = 2.0 cm. (E) qRT-PCR analysis of the TuMV RNA levels in the wild-type Col-0 and the transgenic plants overexpressing AtVSR4 mutants at 14 dpi. *Actin II* was used as an internal control. Error bars represent the standard deviation of three biological replicates of a representative experiment. Statistical analysis was performed using Student's t-test (**, $P < 0.01$; NS, not significant).

signals (1.5–2.5-fold increase) were detected in the leaf samples overexpressing VSR4 or its mutants compared with the control (S5C Fig), and AMPA and DM mutants also showed significantly higher viral CP accumulation levels than VSR4 and AIRAAA. These results suggest

that VSR4 has a pro-viral role in supporting TuMV replication, and also that 6K2 does not hijack the canonical TGN to LE pathway mediated by VSR proteins.

To verify if these VSR4 mutants affected 6K2 distribution, TuMV 6K2 (with an N-terminal yellow fluorescent protein tag) was transiently co-expressed with GUS control or the VSR4 mutants (with a C-terminal myc tag) in *N. benthamiana* leaves. Confocal microscopy showed that in comparison with the control or AIRAAA mutant treatment, transient overexpression of the AMPA or DM remarkably increased the amount of 6K2-induced punctate bodies in the cytoplasm (S6A and S6B Fig). Immunoblotting analysis confirmed that co-expression of the VSR4 mutants did not affect the accumulation levels of 6K2 protein (S6C Fig). Co-IP results revealed that all three VSR4 mutants can still bind with 6K2, although AIRAAA had a relatively weaker binding capacity than wild type VSR4 (S6D Fig).

We further colocalized YFP-6K2 with the endocytic-tracking styryl FM4-64 or cis-Golgi marker Man49-mCherry [49]. Confocal results showed that AMPA and DM treatment increased the chance of 6K2 localizing at cis-Golgi (S7 Fig). The PCC values confirmed our observations (S7 Fig). This result is consistent because AMPA interferes with the anterograde TGN to PVC signal, and so more 6K2 was retained in cis-Golgi. Moreover, these 6K2-induced punctate bodies actually entered into enlarged PVC/MVB/LE, but not granule-like endosomes, labelled by FM4-64 (Figs 4B and S7). We also examined whether AMPA, AIRAAA and DM affect localization at TuMV 6K2-induced vesicles. Confocal results showed that several small vesicles formed by VSR4, AMPA and DM were tightly associated or colocalized with 6K2-induced vesicles, while AIRAAA had fewer small vesicles associated with 6K2 vesicles (Fig 4C).

Finally, to further confirm the effect of VSR4 mutants on TuMV infection, stable transgenic lines carrying the YFP-AtVSR4 mutant fusions under control of the 35S promoter in the *atvsr4* background were generated, and YFP fluorescence was observed by confocal microscopy (S8A Fig). We opted to employ the 35S promoter, because lines under the control of a 1.2-kbp fragment upstream of the *AtVSR4* CDS did not lead to detectable expression. Three independent T0-positive lines of each mutant were identified by immunoblotting (S8B Fig). We selected two homozygous T2 lines for each mutant for TuMV infection assays. In comparison with the wild-type Col-0 plants, the *atvsr4* plants overexpressing AMPA or DM were more susceptible to TuMV infection and developed more severe symptoms (yellowish and dwarf phenotype) at 14 dpi (Fig 4D). Consistently, high levels of viral genomic RNA were also detected in the plants overexpressing APMA and DM (Fig 4E). The *atvsr4* plants overexpressing AIRAAA had similar symptoms and viral accumulation levels to the Col-0 plants. Taken together, the results suggest that VSR4 promotes TuMV replication by promoting formation of 6K2-induced punctate bodies and further targets them into enlarged PVC/MVB/LE via an unconventional pathway.

## AtVSR4 is *N*-glycosylated *in vivo*

Protein trafficking is known to be triggered by post-translational modifications including *N*-glycosylation [54,55]. AtVSR1 was previously identified as an *N*-glycosylated protein, and *N*-glycosylation modification is critical for ligand binding and vacuolar protein trafficking [56]. The NetNGlyc server (http://www.cbs.dtu.dk/services/NetNGlyc/) predicts that AtVSR4 contains three potential *N*-glycosylation sites at residues Asn148, Asn294 and Asn434 in the luminal domain (Fig 5A). To determine whether AtVSR4 is *N*-glycosylated at these predicted residues, we treated *N. benthamiana* leaf tissues transiently expressing AtVSR4-myc with tunicamycin, a known *N*-glycosylation inhibitor, followed by immunoblotting analysis. AtVSR4--myc also only harbors these three predicted *N*-glycosylation sites. We found that AtVSR4-myc

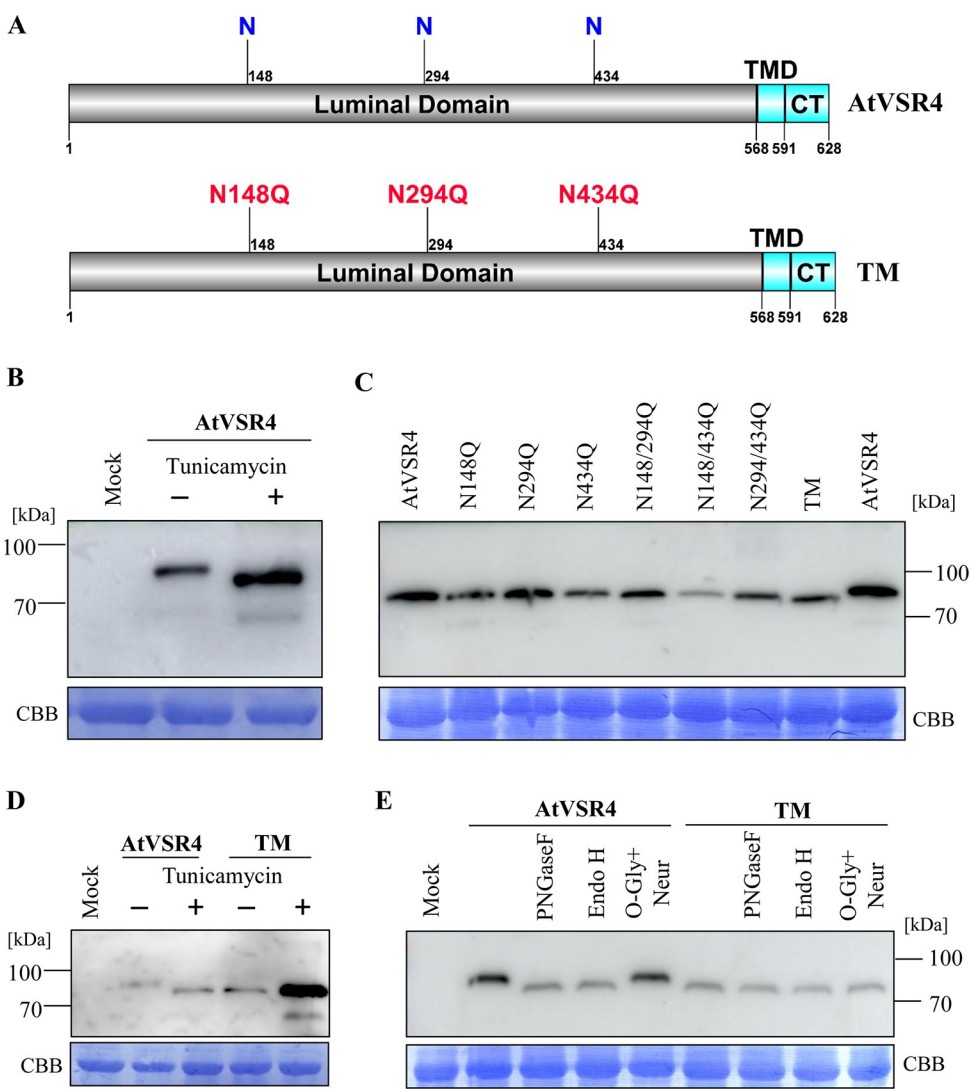

**Fig 5. AtVSR4 is *N*-glycosylated.** (A) Diagram showing the predicted *N*-glycosylated sites N148, N294, N434 in AtVSR4. The triple mutant with amino acid mutations N148Q, N294, and N434Q, designated as TM, is shown. (B) AtVSR4 size migration analysis of protein extracts from *N. benthamiana* cells expressing AtVSR4-c-myc treated with *N*-glycosylation inhibitor tunicamycin or DMSO. (C) AtVSR4 size migration analysis of protein extracts from *N. benthamiana* cells expressing different glycosylation site mutants. (D) AtVSR4 and TM size migration analysis of protein extracts from *N. benthamiana* cells expressing AtVSR4-c-myc or TM treated with tunicamycin or DMSO. (E) AtVSR4 and TM size migration analysis of protein extracts from *N. benthamiana* cells expressing AtVSR4-c-myc or TM treated with different glycosidases. Coomassie Brilliant Blue (CBB) R-250-stained RuBisco large subunit serves as a loading control.

migrated faster in SDS-PAGE extracts from leaf tissues treated with *N*-glycosylation inhibitor tunicamycin than in those from the DMSO-treated controls (Fig 5B), indicating that AtVSR4 indeed harbours *N*-linked oligosaccharides. To analyze which sites are subjected to *N*-glycosylation, each of the potential asparagine (N) residues of AtVSR4-myc was mutated into glutamine (Q), enabling analysis of AtVSR4 with one, two or three of the potential glycosylation sites mutated. These mutants were transiently expressed in *N. benthamiana* leaf cells, followed by Western blot analysis. Mutation on any one or two of the three *N*-glycosylation sites did not result in apparent band shift (Fig 5C). However, mutation of all three sites, the triple

mutant (TM), shifted obviously faster than the wild type AtVSR4-myc (Fig 5C). To exclude the possibility that there might be further *N*-glycosylation sites in addition to those predicted, we further treated the leaf tissues expressing AtVSR4-myc or AtVSR4TM-myc with tunicamycin, followed by Western blot analysis. As shown in Fig 5D, AtVSR4TM-myc had the same molecular weight with or without tunicamycin treatment. AtVSR4-myc protein showed a smaller molecular weight protein band in tunicamycin-treated cells, with the same size as the AtVSR4TM-myc, indicating the absence of *N*-glycosylation in AtVSR4TM-myc. These results demonstrate that these three predicted *N*-glycosylation sites in AtVSR4 harbor *N*-linked glycans.

To analyze the glycosylation pattern of AtVSR4, we treated AtVSR4-myc transiently expressed in *N. benthamiana* cells with various glycosidases, including endoglycosidase H (Endo H) and peptide-N-glycosidase F (PNGase F) [57]. Endo H cleaves high-mannose glycans, which are added to proteins in the ER. Subsequent trimming and addition reactions in the Golgi body yield complex glycans, which are resistant to Endo-H. PNGase F removes most *N*-linked glycans. Following either Endo H or PGNase F treatment, the AtVSR4-myc migrated to the same position as AtVSR4TM-myc (Fig 5E). Moreover, AtVSR4TM-myc protein showed no shift upon treatment with either of these enzymes (Fig 5E), reflecting the size of the fully deglycosylated protein. To exclude the possibility that AtVSR4 and AtVSR4TM may have *O*-glycosylation sites, we further treated AtVSR4 and AtVSR4TM-myc with O-Glycosidase & Neuraminidase Bundle, which can simultaneously remove terminal sialic acid residues and O-linked glycans. Western blot analysis showed no shift change for either AtVSR4-myc or AtVSR4TM-myc (Fig 5E), suggesting that AtVSR4 harbors no *O*-glycosylation sites. These results indicate that AtVSR4 can only be *N*-glycosylated and that it is mature at ER with high-mannose glycans.

## Non-glycosylated AtVSR4 shows a stronger ability to promote TuMV infection

To determine whether the removal of *N*-glycans from AtVSR4 would affect TuMV infection, we first tested the protein-protein interaction between non-glycosylated AtVS4 and 6K2. Both Co-IP and luciferase complementation imaging (LCI) assays showed that non-glycosylated AtVS4 had a weak interaction signal with 6K2, no doubt because there was less protein accumulation of AtVSR4TM in cells, compared with AtVSR4 (Fig 6A and 6B). These results indicate that AtVSR4TM can still interact with 6K2. Then we investigated the effect of transient overexpression of AtVSR4TM on TuMV accumulation. *N. benthamiana* leaves were co-infiltrated with an TuMV::GFP infectious clone and with an AtVSR4TM-myc, AtVSR4-myc or Gus-myc control expression construct. As expected, overexpression of AtVSR4 enhanced the intensity of GFP fluorescence directly resulting from TuMV infection compared with the control treatment (Fig 6C). Interestingly, co-expression of AtVSR4TM resulted in remarkably higher intensity of GFP fluorescence than with AtVSR4 (Fig 6C). Virus accumulation was assessed in infiltrated leaves at 48 and 60 hpai by qRT-PCR and Western blot, respectively. Consistently, we found a significant increase (~2.0 fold) of TuMV genomic (+) and (-) RNA in plants co-infiltrated with AtVSR4TM compared to those with AtVSR4 (Fig 6D). The Western blot assay also showed s substantial increase (~1.7 fold) in TuMV CP accumulation in AtVSR4TM compared with AtVSR4 (Fig 6E).

To further confirm the effect of overexpression of AtVSR4TM on TuMV infection, transgenic Arabidopsis lines overexpressing the fusion proteins Flag-4×myc-AtVSR4 or Flag-4×myc-AtVSR4TM were generated. Eight AtVSR4 and seven AtVSR4TM independent overexpression (oe) lines were examined by immunoblotting to detect the fusion protein (S9 Fig).

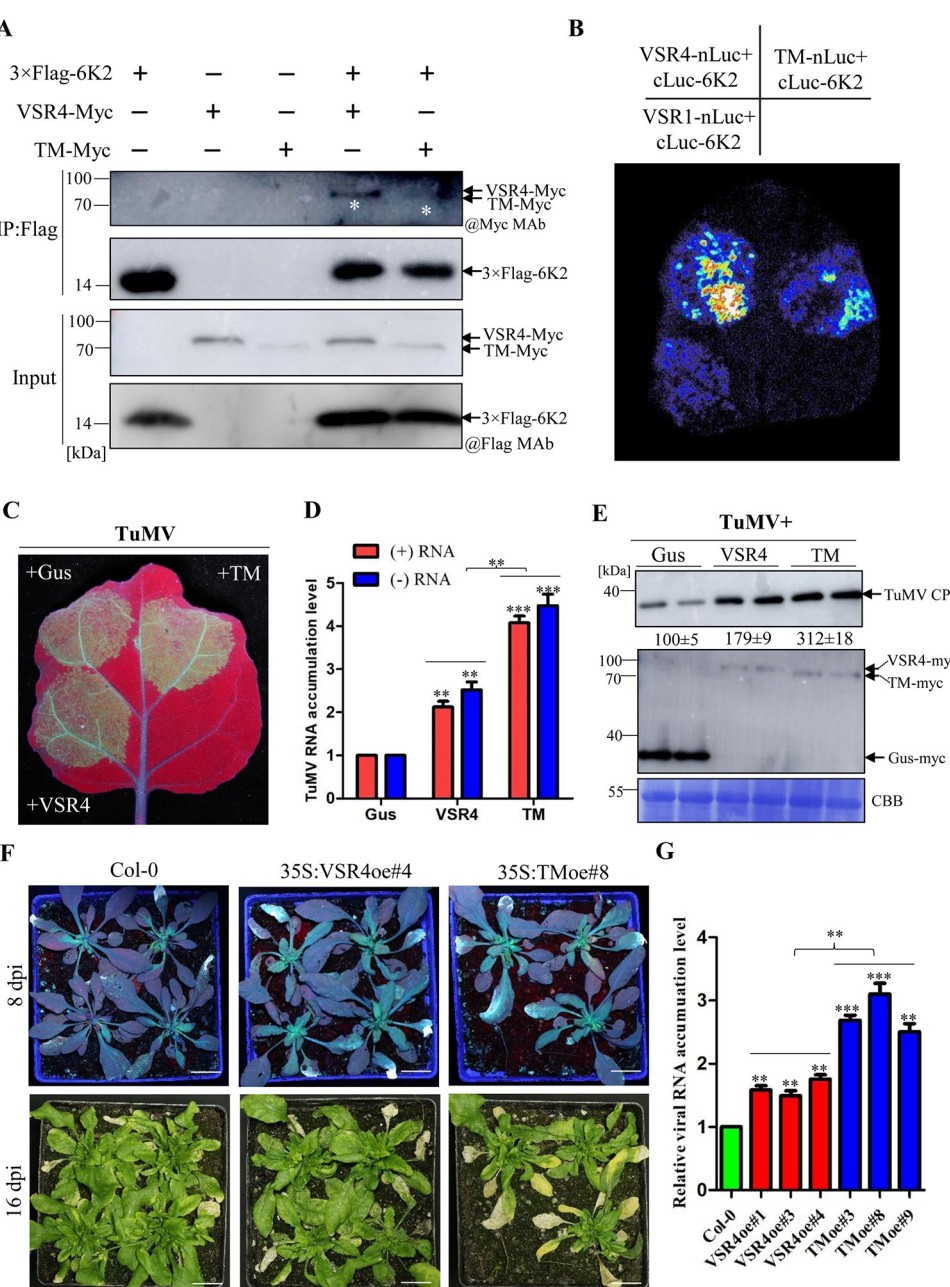

**Fig 6. Effect of non-glycosylated AtVSR4 on its interaction with 6K2 and TuMV infection.** (A) Co-IP assay of the interaction between 6K2 and AtVSR4 or non-glycosylated AtVSR4TM. Different cell lysates were immunoprecipitated with anti-Flag M2 gel beads, separated by SDS-PAGE and immunoblotted with anti-Flag monoclonal antibody (@Flag MAb), or anti-myc monoclonal antibody (@myc MAb). White asterisks denote myc-fusion AtVSR4 or AtVSR4TM bands. (B) luciferase complementation imaging (LCI) assays of interaction between 6K2 and AtVSR4 or AtVSR4TM. C-luc-6K2 and VSR4-N-luc or TM-N-luc were transiently co-expressed in *N. benthamiana*. VSR1-N-luc was used as negative control. (C) GFP fluorescence in *N. benthamiana* plants inoculated with TuMV-GFP together with GUS (control), AtVSR4, or TM. Plants were photographed under a hand-held UV lamp at 3 dpi. (D) Results of qRT-PCR showing the quantification of positive-strand viral genomic RNA [(+)RNA] or negative-strand viral genomic RNA [(-)RNA] accumulation in *N. benthamiana* plants agroinfiltrated with different combinations of plasmids from (C). The infiltrated leaf tissues were collected and pooled at 60 hours post agroinfiltration (hpai) for RNA purification. The purified RNA was analyzed by qRT-PCR with TuMV nib-specific primers using the *actin II* transcript level as an internal control. (E) Immunoblotting analysis of the accumulated TuMV CP levels in the infiltrated leaf tissues from *N. benthamiana* plants in (C) at 60 hpai. The relative TuMV CP signals were quantified by Image J software. Coomassie Brilliant Blue R-250-stained RuBisco large subunit serves as a loading control. TuMV CP was detected with anti-

TuMV CP polyclonal antibody. AtVSR4 and TM was detected with anti-c-Myc monoclonal antibody. (F) Phenotypes of the wild-type Col-0 and transgenic plants overexpressing AtVSR4 or TM after inoculation with TuMV under a hand-held UV lamp at 8 dpi or under regular light at 16 dpi. Scale bar = 2.0 cm. (G) qRT-PCR analysis of the TuMV RNA levels in the wild-type Col-0 and transgenic plants overexpressing AtVSR4 or TM at 16 dpi. *Actin II* was used as an internal control. Data represent means with SD of three independent experiments. Statistical significance was determined by Student's t test (**, $P < 0.01$; ***, $P < 0.001$).

Three T2 lines of each construct (AtVSR4oe#1, #3, #4 and AtVSR4TMoe#3, #8, #9) that expressed relatively higher levels of the fusion proteins were selected for further analysis. Wild-type Col-0, AtVSR4oe, and AtVSR4TMoe Arabidopsis plants were mechanically inoculated using sap from *N. benthamiana* leaves infected by TuMV::GFP. At 8 dpi, both AtVSR4T-Moe and AtVSR4oe showed a remarkably higher intensity of GFP fluorescence than Col-0 plants (Fig 6F). At 16 dpi, symptoms were more severe in AtVSR4TMoe than in AtVSR4oe and Col-0 plants (Fig 6F). qRT-PCR results showed that there were significantly higher levels of TuMV RNA in AtVSR4TMoe than in AtVSR4oe plants (Fig 6G). AtVSR4oe plants also had significantly higher viral RNA accumulation than Col-0 plants. Taken together, these results indicate that non-glycosylated AtVSR4 promotes TuMV infection.

## Non-glycosylated AtVSR4 promotes TuMV 6K2 to form punctate bodies and target enlarged LE

We next wanted to know the mechanism whereby non-glycosylated AtVSR4 promotes TuMV infection. We therefore first investigated the effect of overexpressing AtVSR4TM on 6K2 distribution in *N. benthamiana* leaves. As shown in Fig 7A and 7B, AtVSR4TM overexpression significantly increased the formation of 6K2 punctate bodies at 2 dpai, compared with co-expression of AtVSR4 or expression of 6K2 alone. Immunoblotting analysis confirmed that co-expression of AtVSR4 or AtVSR4TM did not affect the protein accumulation level of YFP-6K2 (Fig 7C). Moreover, the formation of 6K2 punctate bodies apparently decreased (S10 Fig), and there was little expression of AtVSR4 and AtVSR4TM at 3 dpai (Fig 7C), suggesting a tight relationship between the formation of 6K2 punctate bodies and accumulation levels of AtVSR4 or AtVSR4TM protein.

To verify the distribution of these 6K2-induced punctate bodies under AtVSR4TM treatment, we colocalized them with either FM4-64 or cis-Golgi marker. As expected, these punctate bodies did not enter into FM4-64 labeled granule-like endosomes when 6K2 was expressed alone or with AtVSR4 or TM (S11A Fig). However, we observed that punctate bodies induced by 6K2 colocalized with enlarged LE labelled by FM4-64 (Fig 7D), and also with cis-Golgi marker Man49-mCherry when AtVSR4 or AtVSR4TM were overexpressed (S11B and S11C Fig). These results indicate that non-glycosylated AtVSR4 can promote formation of 6K2-induced punctate bodies from cis-Golgi, which then target the enlarged LE.

## TuMV hijacks *N*-glycosylated AtVSR4 and relieves AtVSR4 degradation

Finally, we questioned whether TuMV infection can regulate the *N*-glycosylation levels of AtVSR4. To exclude the possibility that the protein accumulation levels might influence its *N*-glycosylation level, we first detected and compared the amounts of AtVSR4-myc and AtVSR4TM-myc transiently expressed in *N. benthamiana* cells. As shown in Fig 8A, there was very much less accumulation of AtVSR4TM-myc than of AtVSR4-myc at 2 dpai, and both proteins were at levels below detection from 3 dpai. The mRNA levels of *AtVSR4* and *AtVSR4TM* were similar in infiltrated *N. benthamiana* leaves (S12A Fig). We also co-expressed AtVSR4 or AtVSR4TM with a well-known post-transcriptional gene silencing suppressor TBSV P19 and

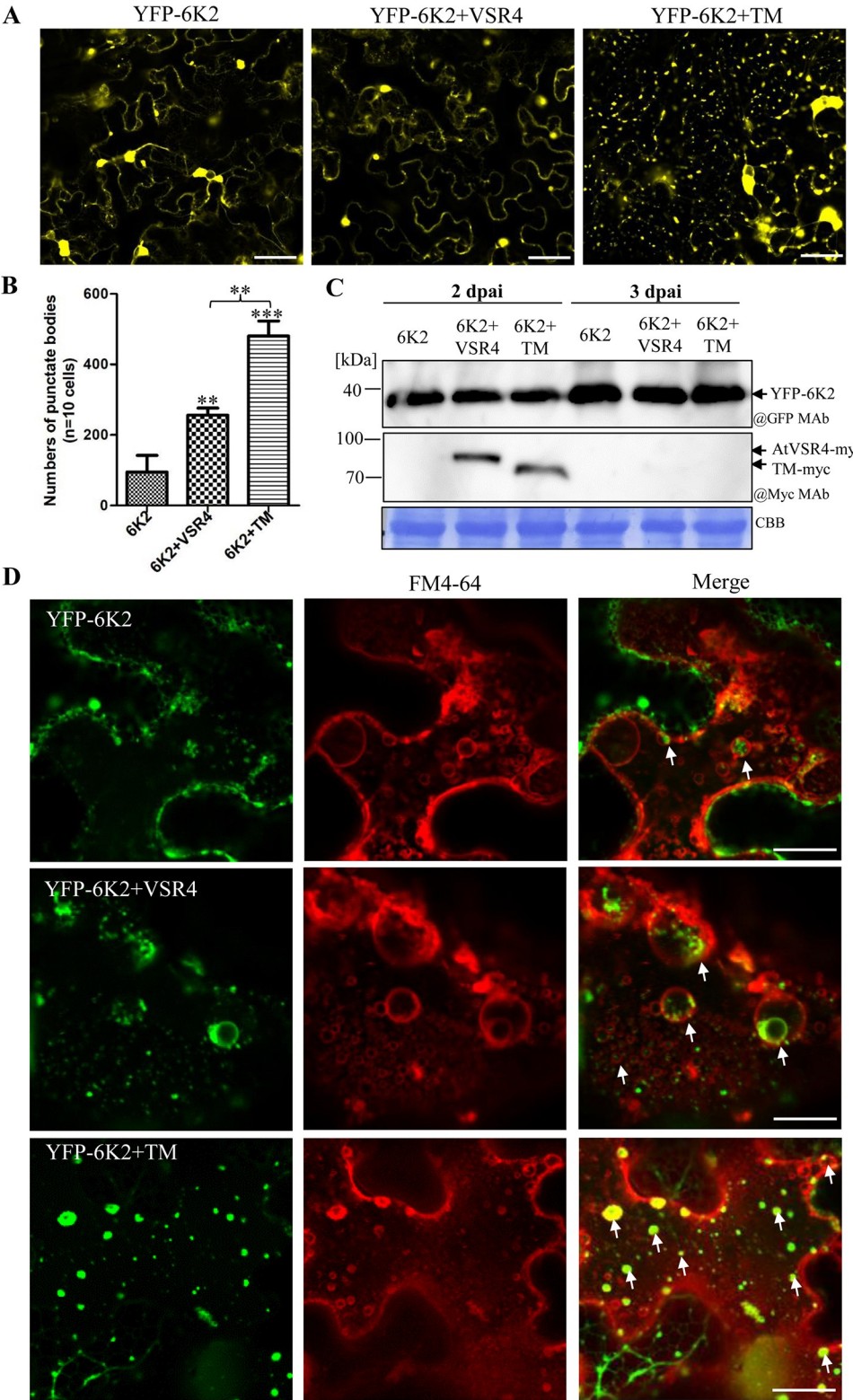

**Fig 7. Effect of non-glycosylated AtVSR4 on subcellular localization of 6K2.** (A) Subcellular localization of 6K2 in *N. benthamiana* cells when expressed alone and when co-expressed with VSR4 or TM. Pictures were taken at 2 dpa. Scale bar represents 20 μm. (B) Number of 6K2-induced punctate bodies in the cytoplasm when YFP-6K2 was expressed alone or when VSR4-myc and TM-myc were co-expressed (10 cells per construct were investigated at 2 dpai

and the number was calculated using Image J software). Values represent the mean number of punctate bodies ±SD per 10 cells from three independent experiments. Statistical significance was determined by Student's $t$ test (**, $P < 0.01$; ***, $P < 0.001$). (C) Immunoblotting of total protein extracts from the *N. benthamiana* leaves treated in (A). The membrane was probed with anti-GFP monoclonal antibody (@GFP MAb), or Myc MAb (@Myc MAb). (D) Subcellular localization of punctate bodies induced by 6K2-induced when VSR4 or TM were overexpressed in *N. benthamiana* leaves. The YFP-6K2 signals were stained with FM4-64. White arrows show the enrichment of YFP-6K2 in FM4-64 labelled single membrane vesicle-like structures (SMVLs). Scale bar represents 20 μm. BF, bright field.

compared their protein accumulation levels in *N. benthamiana* leaves at 2 dpai. Western blot results revealed that P19 treatment indeed increased AtVSR4 and AtVSR4TM accumulation compared with empty vector control treatment but AtVSR4TM still accumulated much less than AtVSR4 (Fig 8B). Taken together, these results suggest that *N*-glycosylation is required for the efficient accumulation of AtVSR4 protein.

Since robust TuMV replication usually occurs at around 2 to 3 dpai and viral intercellular movement can be observed at around 4 dpai [53,58], we monitored the protein levels of AtVSR4 and its non-glycosylated mutant AtVSR4TM under TuMV infection at three time points (2, 3 and 4 dpai). As shown in Fig 8A, the levels of both AtVSR4-myc and AtVSR4TM-myc protein were greater under TuMV infection than in non-infected controls at 2 dpai. The mRNA levels of *AtVSR4* and *AtVSR4TM* in infiltrated leaves were not altered by TuMV infection at this time point (S12B Fig), suggesting that TuMV infection upregulates AtVSR4 protein accumulation, which likely does not occur by increasing its mRNA level. To test this idea, we investigated the endogenous *AtVSR4* level in Arabidopsis Col-0 leaves under TuMV infection. RT-qPCR results showed that the mRNA levels of *AtVSR4* in TuMV-infected leaves were not significantly different to those in mock-treated plants at 4, 7 or 14 dpi (Fig 8C). Western blot analysis of plants with continuous over-expression of AtVSR4 (AtVSR4oe#4) also showed apparently increased AtVSR4 accumulation under TuMV infection compared with the controls at two time points (4 and 7 dpi) (Fig 8D). TuMV encodes two viral silencing suppressors [24]. As expected, the mRNA levels of *AtVSR4* and *AtVSR4TM* in infiltrated leaves under TuMV infection were significantly more than those in mock-treated leaves at 3 and 4 dpai (S12B Fig). Moreover, the mRNA level of AtVSR4 was similar to that of AtVSR4TM under the same treatment. However, glycosylated AtVSR4 bands can be readily detected until 4 dpai, whereas only a weak non-glycosylated AtVSR4TM band can be detected at 3 dpai (Fig 8A), under TuMV infection. This result is consistent with P19 treatment shown in Fig 8B. Notably, we did not observe apparent non-glycosylated bands for AtVSR4 expression treatment under TuMV infection until 4 dpai. Collectively, these results indicate that TuMV increases AtVSR4 accumulation and hijacks glycosylated AtVSR4 for infection.

Finally, to explore how TuMV increases AtVSR4 accumulation, treatment with either carbobenzoxy-L-leucyl-L-leucyl-L-leucinal (MG132) or 3-methyladenine (3-MA), drugs that specifically inhibit the 20S proteasome and autophagy pathways [59] respectively, was performed to test whether these two pathways are involved in AtVSR4 accumulation. Consistent with data shown in Fig 8A, neither AtVSR4 nor AtVSR4TM could be detected at 70 hpai under DMSO treatment (Fig 8E). However, 3-MA treatment partially rescued the accumulation of AtVSR4 and AtVSR4TM, although not to the same extent as that under TuMV infection. Compared with the controls, 3-MA treatment greatly increased AtVSR4TM accumulation under TuMV infection. MG132 treatment gave similar results to those of the DMSO controls. Taken together, the results indicate that TuMV hijacks *N*-glycosylated AtVSR4 and slows its degradation by the autophagy pathway to establish optimal viral infection.

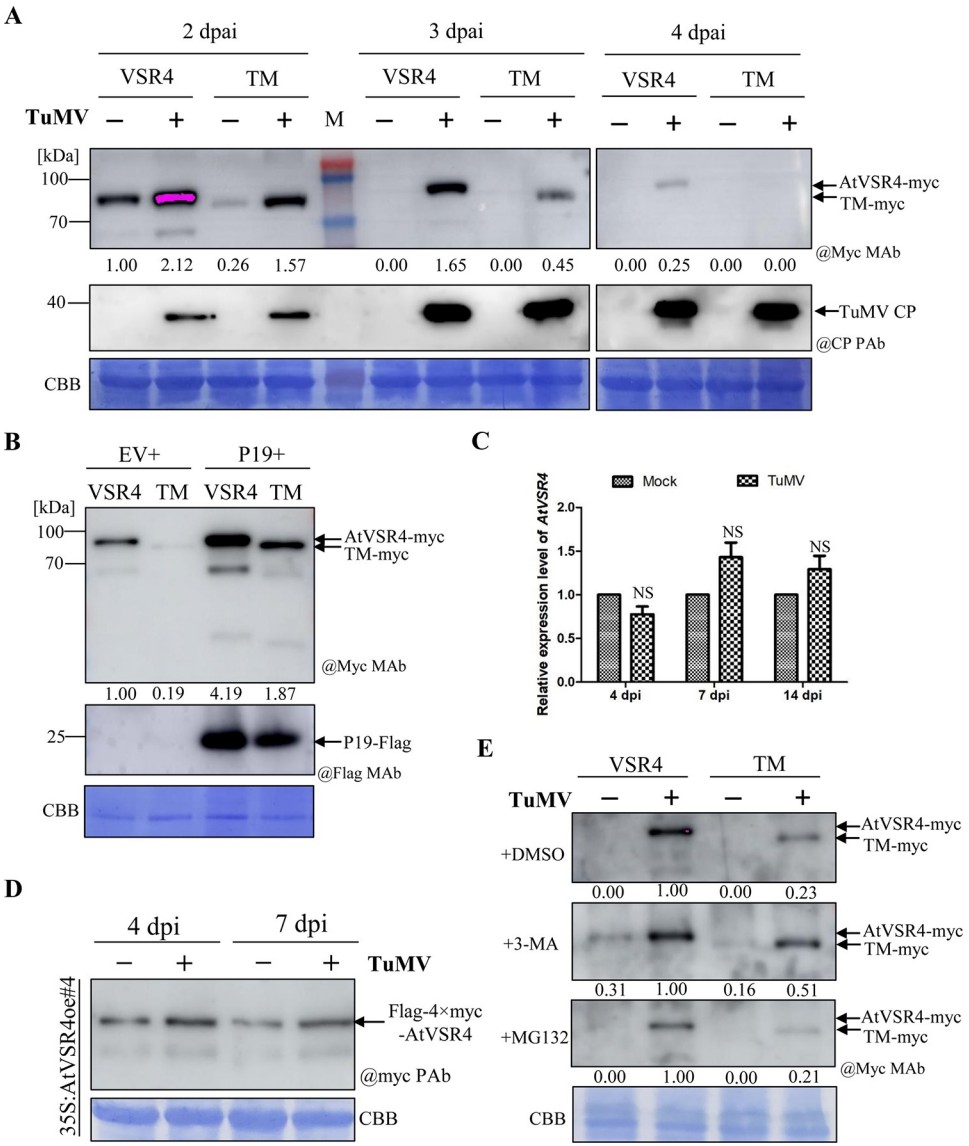

**Fig 8. TuMV hijacks N-glycosylated AtVSR4 and relieves AtVSR4 degradation.** (A) Immunoblots of AtVSR4 or TM accumulation levels in *N. benthamiana* leaves at 2, 3 and 4 d after infection with TuMV and in the corresponding controls. The relative AtVSR4 or TM signals were quantified by ImageJ software. The membrane was probed with anti-myc monoclonal antibody (@Myc MAb), or TuMV CP PAb (@CP PAb). (B) Immunoblots of AtVSR4 and AtVSR4TM accumulation levels when co-expressed with TBSV P19 or the empty vector control at 2 dpai in *N. benthamiana* leaves. (C) RT-qPCR analysis of *AtVSR4* expression in Col-0 leaves under mock treatment and TuMV infection at 4, 7 and 14 dpi. Error bars represent the standard deviation (SD) of three experiments. Statistical analysis was performed using Student's t test (NS, not significant). (D) Immunoblotting analysis showing the accumulation levels of flag-4xmyc-AtVSR4 in AtVSR4oe#4 plants under mock inoculation and TuMV infection at 4 and 7 dpi. Coomassie Brilliant Blue R-250-stained RuBisco large subunit serves as a loading control. (E) The effect of the autophagy inhibitor 3-MA and the 20S proteasome inhibitor MG132 on AtVSR4 or AtVSR4TM accumulation with or without TuMV infection. AtVSR4-myc and AtVSR4TM-myc were transiently expressed alone or co-expressed with TuMV::GFP infectious clone in *N. benthamiana* leaves by Agrobacterium infiltration. At 54 hpi, the infiltrated leaf areas were treated with 10 mM 3-MA or 100 μM MG132 for an additional 16 h. Total proteins from infiltrated regions were extracted and subjected to immunoblotting analysis using anti-c-myc monoclonal antibody (@c-myc MAb) as the primary antibody. CBB, Coomassie brilliant blue-stained gel used as a loading control.

## Discussion

One of the emerging themes in RNA virus replication is that (+) RNA viruses hijack cellular pathways and co-opt host proteins to build large compartments for VRC assembly. However, the actual mechanism of VRC assembly is incompletely understood. The potyvirus TuMV is the only plant virus known to utilize vesicles for intercellular movement [11,60], although virion formation is also required [58,61]. The TuMV-encoded 6K2 protein induces formation of these vesicles, which are vehicles for both intracellular and intercellular movement. Recent reports suggest that 6K2-induced vesicles can utilize an unconventional Golgi-bypassing pathway to target PVC/MVB/LE for systemic infection [10,30]. Almost at the same time, our group also found that TuMV 6K2-induced VRC is associated with PVC/MVB/LE [9] but the mechanism involved is largely unknown. This study points to the involvement of an unconventional VSR-mediated trafficking pathway in TuMV infection. Using a forward genetics system based on available *AtVSR* mutants, we successfully identified *AtVSR4* as a novel proviral host factor for TuMV infection (Figs 1 and 6).

There was no previous information about the function of the VSR protein during plant viral infection, but a protein-protein interaction screen between each of the TuMV-encoded proteins and AtVSR4 identified the small 6K2 as the only viral protein that consistently interacted with AtVSR4 (Fig 2). It was already known that 6K2 and VSR copurify with a PVC/MVB/LE localizing SNARE protein VTI11 [10,46], and we therefore investigated whether VSR4 regulates 6K2 intracellular trafficking to PVC/MVB/LE. Confocal microscopy revealed that VSR4 colocalizes with 6K2-induced vesicles and is recruited into TuMV 6K2-induced VRC (Fig 3C and 3D). Transient overexpression of a VSR4 mutant that can not interact with 6K2 did not show enhanced TuMV infection (S3 Fig), and knockout of *AtVSR4* interfered with targeting of 6K2 to enlarged LE (Fig 3E and 3F). These results clearly show that the AtVSR4-6K2 interaction is critical for TuMV infection, likely via mediating 6K2 to target enlarged LE. Further site-directed mutagenesis of VSR4 and a TuMV infection assay revealed that AMPA and DM mutants remarkably enhanced TuMV replication (S5 Fig). Because the YMPL motif is required for recognition by the μ1-Adaptin of the adaptor protein (AP) 1 complex and post-Golgi trafficking [40,41], this implied that 6K2 does not utilize the conventional TGN to PVC route. Notably, AMPA and AIRAA mutants still can bind with 6K2 although that of AIRAAA is relatively weaker (S6D Fig). Also, AIRAAA had fewer small vesicles associated with 6K2 vesicles (Fig 4C). These results may explain why the AIRAAA mutant can enhanced TuMV replication when transiently overexpressed in *N. benthamiana* cells (S5 Fig), but not in *atvsr4* plants stably overexpressing AIRAAA (Fig 4D and 4E). The AIRAAA mutant can also affect the endocytosis signal, and the endocytosis pathway has been demonstrated to be required for TuMV infection [8,9], so we cannot exclude the possibility that this also provides an explanation for the results. Overall, these results corroborate the idea that 6K2 does not employ the conventional TGN to PVC route for efficient viral infection [10,31], and explain how accumulation of 6K2-induced VRC or replication proteins in PVC/MVB/LE or the vacuole is beneficial to TuMV infection [9,10,62]. Tombusviruses orchestrate the host endomembrane system and interact with endosomal proteins directly probably providing endosomal lipids for VRC formation [17]. It is possible that AtVSR4-containing LEs target the replication site, and exert a proviral function by a similar mechanism.

Our data indicate that AtVSR4 is modified by the addition of high-mannose glycans at essentially three sites, N148, N294 and N434 (Fig 5). A similar situation was observed in AtVSR1, which also harbors three *N*-glycosylated sites [56]. However, unlike AtVSR4, AtVSR1 contains N-linked complex-type oligosaccharides [56], which are resistant to Endo H. Addition of N-linked complex glycans occurs at Golgi, and therefore AtVSR4 would be in a mature

glycosylated form at the ER, where it binds to 6K2 (Fig 2), consistent with a previous report that VSR binds its ligand at the ER [43]. The 6K2-VSR4 complex then bypasses cis-Golgi and enters into FM4-64 labelled SMVLs (Fig 2). VSR4 overexpression significantly promoted the formation of punctate bodies that are induced by 6K2 (Fig 7A and 7B) and which are enriched in SMVLs (Fig 7D). These SMVLs are very similar to those observed in Fig 4B. Notably, we obtained consistent results showing that enrichment of 6K2 in SMVLs promotes TuMV replication (Figs S5 and 6C–6E). Recent reports suggest that these SMVLs are enlarged PVC/MVB/LE [2,9,10,30], and therefore our results reveal an important role for VSR4 in facilitating 6K2 to target enlarged PVC/MVB/LE by mediating the disassociation of 6K2 from cis-Golgi and entry into an unconventional trafficking pathway. This is not surprising because previous studies have shown that several different endosomal proteins (Rab guanosine triphosphatase (GTPase) family proteins, retromer, and ESCRT proteins) are recruited by different (+) ssRNA viruses to assist in biogenesis of the viral replication compartment, in lipid transport or as intracellular trafficking viral factors to promote virus infection in plants [7,15,17,21,23]. We have recently reported that TuMV replication proteins VPg and CI can hijack endocytosis and post-Golgi pathway to target PVC/MVB/LE for viral replication [8,9], but these act in different ways to 6K2. It therefore appears that the TuMV replication proteins 6K2, VPg, and CI are all transported to PVC/MVB/LE for VRC assembly [9], but they do so by hijacking different transport pathways of the endomembrane system.

In this study, the glycosylation-deficient mutant VSR4TM was expressed relatively poorly compared to the wild type protein (Fig 8A). Many lines of evidence have shown that glycoproteins are more stable than their corresponding non-glycosylated counterparts [63,64]. Thus, non-glycosylated VSR4 is likely subjected to a quality control mechanism that induces its degradation. The non-glycosylated AtVSR4TM dramatically promoted the formation of 6K2-induced punctate bodies and their accumulation in SMVLs (Fig 7). Moreover, consistent with a previous report [56], co-IP and LIC assays revealed that non-glycosylated AtVSR4TM bound much more weakly to cargo protein 6K2 than did wild type AtVSR4 (Fig 6A and 6B), which may be because there was less accumulation of non-glycosylated VSR4. It is possible that overexpression of the non-glycosylated AtVSR4 can compete with endogenous VSRs in the anterograde route from TGN to PVC/MVB/LE, and therefore an increase in the number of endogenous VSRs in the cis-Golgi can be beneficial to TuMV infection. This idea was supported by the observation that more 6K2 punctate bodies apparently localized at cis-Golgi when AtVSR4 or AtVSR4TM were co-expressed as compared with 6K2 expression alone (S11C Fig). Whether over-accumulation of VSR4 and its mutants in cis-Golgi can cause Golgi stress and further stimulate 6K2 body formation and TuMV replication needs to be further investigated. TuMV infection upregulates VSR4 accumulation (Fig 8A) and likely recruits VSR4 to VRC to avoid degradation by autophagy (Figs 3C, 3D and 8A–8C). As obligate parasites, viruses must establish an intimate relationship with their hosts in order to infect, replicate, and disseminate [65]. Although non-glycosylated VSR4 can promote TuMV infection more efficiently than glycosylated VSR4 (Fig 5), we did not observe decreased levels of VSR4 N-glycosylation during TuMV infection (Fig 8). Since N-glycosylation of VSR plays important roles in plant growth [36,56], and non-glycosylated VSR4 is unstable (Fig 8), it appears that TuMV has utilized N-glycosylated VSR4 to establish its relationship with host plants during the evolutionary arms race.

Based on our data and the discussion above, we propose a model to summarise how TuMV co-opts VSR4 to facilitate viral infection (Fig 9). Upon translation of the viral genome at the ER, TuMV 6K2 is recognized by mature N-glycosylated VSR4. The VSR4-6K2 complex is then transported in COPII-coated vesicles to cis-Golgi. A fraction of the 6K2-VSR4 complex disassociates, 6K2 directly targets chloroplasts from cis-Golgi for the formation of replication

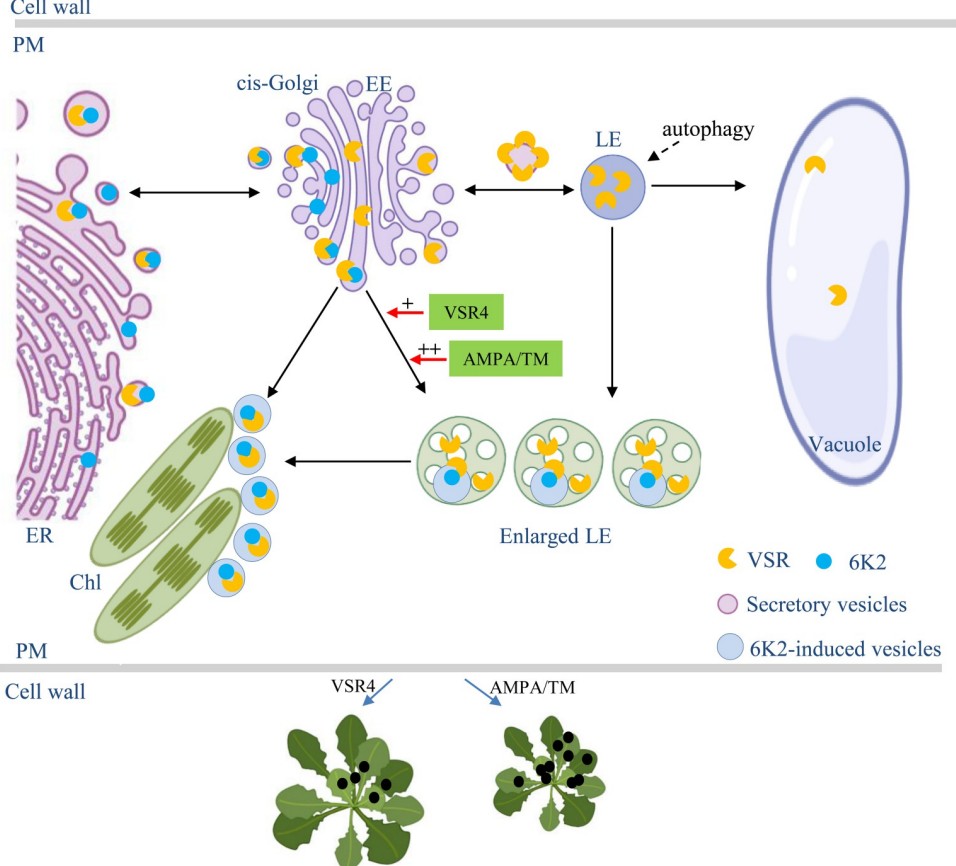

**Fig 9. Proposed model of the role of the VSR protein and its non-glycosylated mutant in TuMV 6K2 intracellular trafficking and viral infection.** When viral proteins are synthesised at the ER, TuMV 6K2 is recognized by VSR4. The VSR4-6K2 complex is then transported in COPII-coated vesicles to cis-Golgi. Some of the VSR4 disassociates from 6K2, enters into endosomal compartments (EE and LE) and finally into vacuoles for degradation by the autophagy pathway. Some of the 6K2 disassociates from cis-Golgi and targets chloroplasts to form replication vesicles. The remaining VSR4-6K2 complex dis-associates from cis-Golgi and targets enlarged LE and chloroplasts via an unconventional pathway for robust replication. The AMPA and non-glycosylated VSR4 mutant TM can promote this progress and viral symptom development. The figure was partially created using BioRender. EE, early endosome. LE, late endosome. ER, Endoplasmic reticulum. PM, plasma membrane. Chl, Chloroplast.

vesicles [28], while VSR4 enters into conventional endosomal compartments, which may lead to VSR4 degradation by the autophagy pathway. The remaining VSR4-6K2 complex can disassociate from cis-Golgi and target enlarged PVC/MVB/LE and chloroplasts for robust virus replication. The AMPA and non-glycosylated mutant TM can promote this progress and viral infection more efficiently compared with VSR4. Ultimately, TuMV co-opts the *N*-glycosylated VSR4 to achieve optimal infection.

## Materials and methods

### Plant materials and growth condition

*Nicotiana benthamiana* and Arabidopsis plants were grown in pots at 23˚C under a photoperiod of 14 h light/10 h dark and 60% humidity. Wild type Arabidopsis ecotype Col-0, and *atvsr* mutants used in this study were obtained from the AraShare (a non-profit Arabidopsis share center, https://www.arashare.cn/index/). Homozygous T-DNA insertion was verified by PCR

as described previously [8]. Transgenic wild type Arabidopsis (ecotype Col-0) or *atvsr4* plants overexpressing *AtVSR4* or its mutants were obtained by the floral-dip method [66]. Transgenic dsRNA reporter *N. benthamiana* B2:GFP seeds were kindly provided by Christophe Ritzenthaler (Centre National de la Recherche Scientifique, France).

## RNA extraction and RT-qPCR

Total RNA was extracted by Trizol following the manufacturer's instructions (Invitrogen). After genomic DNA removal and verification of RNA quality and purity, total RNA extracts were processed for RT as indicated by the manufacturer (SuperScript™ III Reverse Transcriptase, Thermo Fisher). qPCR was performed with the SYBR Green PCR master mix kit (Vazyme, http://www.vazymebiotech.com/), on a LightCycler 480 II real-time PCR system thermal cycler (Roche, Germany), and data analysis was done with LightCycler 480 manager software (Roche). qPCR primer specificity was validated (melt curve) from pooled cDNA samples. The reference gene *actin II* [67] was selected for both *N. benthamiana* and *A. thaliana*. All primers used in this study are listed in the S1 Table.

## Gene cloning and plasmid construction

PrimerSTAR GXL DNA Polymerase (Takara, Japan) was used to amplify all DNA sequences, and Gateway technology (Thermo Fisher Scientific, USA) was employed for plasmid construction. The coding sequence of *AtVSR4* was amplified from *A. thaliana* Col-0 cDNA as described above. Coding sequences of TuMV genes were amplified from the infectious clone TuMV:: GFP deposited in our lab [68]. These cloned genes were recombined into pDONR221 prior to final recombination to different plant expression vectors by LR reactions as previously described [8]. Mutagenesis was performed using overlapping PCR or the QuickChange II XL site-directed mutagenesis kit (Agilent), according to the manufacturer's instructions. All constructs were verified by DNA sequencing.

## Total protein extraction and immunoblotting

*N. benthamiana* leaf tissues were sampled with a hole punch (1.0 cm in diameter). Three leaf disks from different leaves were ground into fine powder in liquid nitrogen, mixed with 100 μl protein extraction buffer and 25 μl 5 × SDS-PAGE loading buffer, and vortexed for 1 min. After being boiled at 75˚C for 15 min, the total protein lysate was quick-chilled on ice for 2 min, centrifuged at 12, 000 rpm for 2 min at 4˚C, and stored at -40˚C until use. Chemiluminescent imaging was performed using an Immobilon Western chemiluminescent horseradish peroxidase (HRP) substrate (Millipore) on an Amersham Imager 680 machine (GE, America) according to the manufacturer's instructions. In each experiment, a parallel gel was stained with Coomassie brilliant blue (CBB) to monitor equal loading of the samples.

## Protein-protein interaction analysis

Membrane yeast two-hybrid tests were performed using the DUALmembrane system (Dualsystems Biotech AG, Switzerland) as previously described [8]. Agrobacterium-mediated transient protein expression was done essentially as previously described [9]. Three to four week-old *N. benthamiana* plants were used for agroinfiltration. For BiFC, the TuMV and host genes were fused with p35S-gateway-YFP-C and p35S-YFP-N [69] respectively, and the $OD_{600}$ was adjusted to 0.4–0.5 for each gene construct. For subcellular localization assays, the $OD_{600}$ was adjusted to 0.1 and for TuMV agro-infiltration assays it was adjusted to 0.05–0.1 unless otherwise specified. For all co-expression combinations, equal volumes of Agrobacterium

suspension were mixed thoroughly prior to agroinfiltration. To compare the protein or viral accumulation levels under different treatments, Agrobacterium suspensions were infiltrated into the same leaf unless otherwise specified. Co-immunopurification experiments were carried out with anti-flag M2 gel (Sigma) as previously described [8]. Briefly, *N. benthamiana* leaves expressing combinations of different proteins were harvested and total proteins were extracted. The cleared lysates (input) were immunopurified using anti-Flag M2 gel, followed by immunoblot analysis of input and immunopurified fractions using specific antibodies.

LCI assays were performed as previously reported [70]. Briefly, the protein coding regions were ligated into pCAMBIA1300-cLUC or pCAMBIA1300-cLUC vectors. The recombinant constructs were agroinfiltrated into different areas of the same *N. benthamiana* leaf with a final concentration of $OD_{600} = 0.50$. The 0.2 mM LUC substrate was then infiltrated into the same areas and imaged at 2 dpi using a low-light cooled CDD imaging system (Amersham Imager 680, GE, America).

## Confocal microscopy and FM4-64 staining

*N. benthamiana* leaves were infiltrated with 20 μM FM4-64 (Invitrogen) and then transferred to the microscope for imaging as previously described [8,71]. The agroinfiltrated leaf tissues of *N. benthamiana* leaves were observed at 48 to 72 h post inoculation unless otherwise specified on a Nikon A1R HD25 confocal microscope (Nikon Microsystems, Japan) [9]. Fluorescence signals for GFP (488 nm/496–518 nm), YFP (514 nm/529–550 nm), and mRFP (543 nm/593–636 nm) were detected. Sequential scanning was used to avoid any interference between fluorescence channels. When YFP was co-imaged with mRFP, excitation was performed with the argon laser at 488 nm and emission recorded at 500–540 nm. The sequential scanning mode was applied for co-imaging of different fluorescent proteins. Image processing was performed with the NIS-elements viewer 4 software (Nikon Microsystem).

## Glycosidase treatment

The N-glycosylation sites of AtVSR4 and c-myc tag amino acid sequences were predicted with NetNGlyc 1.0 online software (http://www.cbs.dtu.dk/services/NetNGlyc/). AtVSR4 and its mutants were transiently expressed in *N. benthamiana* leaves, and then total proteins were extracted and subjected to glycosidase treatment. The peptide N-glycosidase F (PNGase F; NEB), Endo H (NEB) and O-Glycosidase & Neuraminidase Bundle treatments were carried out according to the manufacturer's instructions. In brief, the protein samples in 1× glycoprotein denaturing buffer were heated at 95°C for 10 min and then cooled at room temperature for 5 min. After a short spin, the PNGaseF, Endo H and O-Glycosidase & Neuraminidase Bundle reaction was performed.

## Chemical treatment

3-MA (Sigma) and MG-132 (Sigma) were used to inhibit the autophagy and 26S proteasome pathways, respectively, as previously reported [72].

## Protoplast isolation and plasmid transfection

Mesophyll protoplasts were prepared from 4 week-old *Arabidopsis* leaves by the procedure described previously [73]. About $1 \times 10^4$ protoplasts were transfected with 5 μg plasmids for each subcellular localization assay [9].

## Data analysis

ImageJ was used to quantify average integrated density values of bands on immunoblots and to calculate the numbers of punctate bodies in confocal images as the instructed in the manual. Experiments were repeated at least three times unless otherwise specified. Statistical significance was analyzed using Student's *t*-test. Significance values with $P < 0.05$, 0.01 or 0.001 were denoted *, **, ***, respectively. All analyses were performed using GraphPad Prism software. The working model diagram in Fig 9 was constructed using the website https://biorender.com/.

## Accession numbers

Sequence data from this article can be found in GenBank under the following accession numbers: AT3G52850 (VSR1), AT2G30290 (VSR2), AT2G14740 (VSR3), AT2G14720 (VSR4), AT2G34940 (VSR5), AT1G30900 (VSR6), AT4G20110 (VSR7), AT3G18780 (Actin II), AY179605 (NbActin), and NC_002509 (TuMV strain UK1).

## Supporting information

**S1 Table. Primers used in this study.**
(DOCX)

**S1 Fig. Characterization of *atvsr* T-DNA mutants.** (A) Schematic diagram of the *AtVSRs* gene structure. The relative position of the T-DNA insertion site is shown. Exons are represented by black boxes. (B) PCR genotyping of the *atvsr* mutants. (C) RT-qPCR assay of the mRNA expression level of *AtVSRs* in the corresponding mutants.
(TIF)

**S2 Fig. BiFC assay of negative controls.** Endosome-localized AtVSR1 and TuMV-encoded membrane protein 6K1 are included as negative controls. Scale bar = 20 μm.
(TIF)

**S3 Fig. AtVSR4-6K2 interaction is critical for TuMV infection.** (A) Schematic representation of the functional domains of AtVSR4. Luminal domain, the N-terminal luminal binding domain; TMD, the transmembrane domain; CT, the C-terminus cytoplasmic tail. (B) Protein-protein interactions between each domain or truncated mutants in CT and TuMV 6K2 were examined *in planta* by BiFC. Interaction assays were performed in epidermal cells of *N. benthamiana* leaves. Reconstructed YFP signals were observed at 2 dpai. Scale bar = 20 μm. (C) Results of co-IP assay showing that AtVSR4-C1A cannot form complexes with TuMV 6K2 in *N. benthamiana* cells. Different cell lysates were immunoprecipitated with anti-Flag M2 gel beads, separated by SDS-PAGE and immunoblotted with anti-Flag monoclonal antibody (@Flag MAb), or anti-GFP monoclonal antibody (@GFP MAb). (D) GFP fluorescence in plants inoculated with TuMV-GFP together with GUS (control), AtVSR4, or AtVSR4-C1A. Plants were photographed under a hand-held UV lamp at 3 dpai. **(E)**, Results of qRT-PCR to quantify the levels of positive-strand viral genomic RNA [(+)RNA] and negative-strand viral genomic RNA [(-)RNA] in *N. benthamiana* plants agroinfiltrated with different combinations of plasmids from (D). Statistical analysis was performed using Student's t test (**, $P < 0.01$; NS, not significant). **(F)** Immunoblotting analysis of the accumulated TuMV CP levels in the infiltrated leaf tissues from *N. benthamiana* plants in (D) at 3 dpai.
(TIF)

**S4 Fig. Subcellular localization of AtVSR4 in *N. benthamiana* cells.** (A) AtVSR4 was fused with either GFP at the C-terminus or with YFP at the N-terminus. (B) Colocalization of

AtVSR4 with a cis-Golgi marker Man49-mCherry (upper channels) or an ER marker mCherry-HDEL (lower channels). (C) Subcellular localization of RFP-AtVSR4 in 35S:B2-GFP *N. benthamiana* plants. Photos were taken at 2 dpai. Scale bar = 20 μm. BF, bright field.
(TIF)

**S5 Fig. Effect of AtVSR4 mutants on TuMV infection in *N. benthamiana* cells.** (A) GFP fluorescence in plants inoculated with TuMV-GFP together with GUS (control), AtVSR4, or its mutants. Plants were photographed under a hand-held UV lamp at 3 dpai. (B) Results of qRT-PCR to quantify the levels of positive-strand viral genomic RNA [(+)RNA] and negative-strand viral genomic RNA [(-)RNA] in *N. benthamiana* plants agroinfiltrated with different combinations of plasmids from (A). Statistical analysis was performed using Student's t test (***, $P < 0.001$; **, $P < 0.01$; NS, not significant). (C) Immunoblotting analysis of the accumulated TuMV CP levels in the infiltrated leaf tissues from *N. benthamiana* plants in (A) at 72 hpai.
(TIF)

**S6 Fig. Effects of overexpression of AtVSR4 mutants on TuMV 6K2 distribution in *N. benthamiana* cells.** (A) Subcellular localization of YFP-6K2 when co-expressed with GUS control or with AtVSR4 mutants fused with a c-myc tag at the C-terminus in *N. benthamiana* cells. Scale bar = 20 μm. (B) Number of 6K2-induced punctate bodies in the cytoplasm when YFP-CI was co-expressed with AtVSR4 mutants or with the GUS control (10 cells per construct were investigated at 2 dpai and the number was calculated using Image J software). Values represent the mean number of punctate bodies ±SD per 10 cells from three independent experiments. Statistical analysis was performed using Student's t test (**, $P < 0.01$; NS, not significant). (C) Immunoblotting analysis of the expression of YFP-6K2 and AtVSR4 mutants from A in *N. benthamiana* cells at 2 dpai. Coomassie Brilliant Blue R-250-stained RuBisco large subunit serves as a loading control. 6K2 and AtVSR4 mutants or GUS were detected with anti-GFP and anti-c-myc monoclonal antibodies, respectively. (D) co-IP assay of protein-protein interaction between 6K2 and each of the VSR4 mutants in *N. benthamiana* cells. Different cell lysates were immunoprecipitated with anti-Flag M2 gel beads, separated by SDS-PAGE and immunoblotted with anti-Flag monoclonal antibody (@Flag MAb), or anti-GFP monoclonal antibody (@GFP MAb).
(TIF)

**S7 Fig. Subcellular localization of the YFP-6K2 when co-expressed with AtVSR4 mutants in *N. benthamiana* cells.** (A) colocalization of the YFP-6K2 with FM4-64. (B) colocalization of the YFP-6K2 with cis-Golgi marker (Man49-mCherry). Pictures were taken at 2 dpai. Scale bar = 20 μm. (C) Pearson's correlation coefficient (PCC) values quantifying the colocalization between 6K2 and cis-Golgi marker under different treatments. PCC was measured from 40 cells. Error bars represent the standard deviations (SD) from three experiments. Statistical analysis was performed using Student's t test (**, $P < 0.01$; NS, not significant).
(TIF)

**S8 Fig. Characterization of the transgenic plants overexpressing *AtVSR4* mutants in the *atvsr4* background.** (A) Confocal examination of YFP expression in transgenic plants overexpressing different *AtVSR4* mutants. Scale bar = 50 or 100 μm. (B) Immunoblotting of three positive T0 independent lines showing correct overexpression for each of three AtVSR4 mutants.
(TIF)

**S9 Fig. Verification of transgenic Arabidopsis lines expressing AtVSR4 (A) or TM (B).**
Total leaf protein extracts were immunoblotted with anti-c-myc polyclonal antibodies.
(TIF)

**S10 Fig. Subcellular localization of 6K2 in *N. benthamiana* cells when expressed alone or when co-expressed with VSR4 or TM.** Pictures were taken at 3 dpai. Scale bar = 20 μm.
(TIF)

**S11 Fig. Co-localization of YFP-6K2 co-expressed with either AtVSR4 or TM, with either FM4-64 or cis-Golgi marker (Man49-mCherry).** (A) Co-localization of YFP-6K2 co-expressed with either AtVSR4 or TM with FM4-64. (B) Co-localization of YFP-6K2 co-expressed with either AtVSR4 or TM with cis-Golgi marker (Man49-mCherry). Pictures were taken at 2 dpai. Scale bar = 20 μm. (C) Pearson's correlation coefficient (PCC) values quantifying the colocalization between 6K2 and cis-Golgi maker under different treatments. PCC was measured from 40 cells. Error bars represent the standard deviations (SD) from three experiments. Statistical analysis was performed using Student's t test (**, $P < 0.01$; NS, not significant).
(TIF)

**S12 Fig. RT-qPCR analysis of mRNA levels of *AtVSR4* and *AtVSR4TM* in non-infected controls (A) and following TuMV infection (B) at 2, 3 and 4 dpai.** Error bars represent the standard deviations (SD) of three experiments. Statistical analysis was performed using Student's t test (**, $P < 0.01$; ***, $P < 0.001$; NS, not significant).
(TIF)

## Acknowledgments

We are indebted to Prof. Christophe Ritzenthaler (Université de Strasbourg, France) for the transgenic T2 dsRNA reporter *N. benthamiana* line B2-GFP, Prof. Wenhua Zhang and Xiaorong Tao (Nanjing Agricultural University, China) for the cis-Golgi construct Man49-mCherry, Prof. Xiaofei Cheng (Northeast Agricultural University, China) for the pBA-Flag-4×c-Myc vector, Prof. Liwen Jiang (The Chinese University of Hong Kong, China) and Jinbo Shen (Zhejiang A&F University, China) for the AtARA7^Q69L clone, and Prof. M. J. Adams (Minehead, UK) for polishing the manuscript.

## Author Contributions

**Conceptualization:** Guanwei Wu, Aiming Wang, Jianping Chen, Fei Yan.

**Data curation:** Guanwei Wu, Zhaoxing Jia, Kaida Ding, Lin Lin, Jiejun Peng, Shaofei Rao.

**Formal analysis:** Guanwei Wu, Hongying Zheng, Yuwen Lu, Lin Lin, Jiejun Peng, Shaofei Rao, Fei Yan.

**Funding acquisition:** Guanwei Wu, Hongying Zheng, Fei Yan.

**Investigation:** Guanwei Wu, Zhaoxing Jia, Hongying Zheng, Yuwen Lu, Lin Lin, Jiejun Peng.

**Methodology:** Guanwei Wu, Kaida Ding, Shaofei Rao.

**Project administration:** Jianping Chen, Fei Yan.

**Resources:** Jianping Chen, Fei Yan.

**Software:** Guanwei Wu.

**Supervision:** Jianping Chen, Fei Yan.

**Validation:** Guanwei Wu.

**Visualization:** Guanwei Wu.

**Writing – original draft:** Guanwei Wu.

**Writing – review & editing:** Guanwei Wu, Aiming Wang, Jianping Chen, Fei Yan.

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
