## [Decision Letter · Decision Letter 0]

28 Dec 2021

Dear Dr. Yan,

Thank you very much for submitting your revised manuscript "Turnip mosaic virus co-opts the vacuolar sorting receptor VSR4 to promote viral genome replication in plants by targeting viral replication vesicles to the endosome" for consideration at PLOS Pathogens. As with all papers reviewed by the journal, your manuscript was reviewed by members of the editorial board and by several independent reviewers. The reviewers appreciated the attention to an important topic. Based on the reviews, we are likely to accept this manuscript for publication, providing that you modify the manuscript according to the review recommendations.

Sincerely,

Savithramma P. Dinesh-Kumar

Associate Editor

PLOS Pathogens

Peter Nagy

Section Editor

PLOS Pathogens

Kasturi Haldar

Editor-in-Chief

PLOS Pathogens

orcid.org/0000-0001-5065-158X

Michael Malim

Editor-in-Chief

PLOS Pathogens

orcid.org/0000-0002-7699-2064

Reviewer Comments (if any, and for reference):

Reviewer's Responses to Questions

**Part I - Summary**

Reviewer #1: This manuscript addresses the identification of AtVSR4 as an interactor of TuMV 6K2 protein and aims to characterize its role in targeting 6K2 to the enlarged late endosome for efficient viral infection instead of using the conventional VSR-mediated pathway from early to late endosome. The authors report VSR4 mutants defective in trafficking and non-N-glycosylated forms that support stronger viral replication than the wild type VSR4. At the same time, the virus promotes VSR4 accumulation and hijack glycosylated VSR4 for its protection from autophagy pathway. The manuscript is original and provides a large amount of information on the role of 6K2 co-opting VSR4 and the way they are able to traffic to promote viral replication. These results are relevant for the plant-virus community. The methodology is adequate. The manuscript is dense to read probably because it has a large amount of information. Overall, it is well written.

The doubts I had referred to mutants and non-N-glycosylated VSR4 promoting viral infection. Then, wild type VSR4 would be somehow down controlling viral infection. However, this question has already been addressed by the authors in response to the previous reviewers and included in the new version. Thus, I have no further requirements.

Reviewer #2: This manuscript continues recent investigations on ER-derived 6K2-induced vesicles and how they utilize an unconventional Golgi-bypassing pathway and enlarged late endosomes to induce a productive turnip mosaic virus infection. The authors show through extensive experimental work that vacuolar sorting receptor 4 (VSR4) has a specific role in this process through its interaction with the 6K2 protein. The authors demonstrate that QYMDS sequence at a C-terminal domain of VSR4 serves as a binding site and when this sequence was substituted with five alanine residues both 6K2 binding and CP accumulation were reduced (Fig. S3). All conditions that increased the amount of VSR4 and those mutations in it which prevented its normal function in Golgi traffic, enhanced 1) the amount of unconventional bypass of 6K2-vesicles from cis-Golgi to enlarged late endosomes and 2) both (+)- and (-)-strand TuMV RNA accumulation. N-glycosylation is used for the turnover regulation of VSR4. TuMV infection increased the amount of VSR4 on protein but not on mRNA level, which proposes that TuMV can affect the degradation of glycosylated VSR. The authors have done a substantial revision and added a lot of data to this revised version. I find the presented data very interesting.

**Part II – Major Issues: Key Experiments Required for Acceptance**

Reviewer #1: (No Response)

Reviewer #2: Among the added data is Fig S3 in which it is shown that QYMDS to AAAAA substitution in C-terminus of VSR inhibited the VSR4 binding to 6K2 and reduced TuMV CP accumulation. The quantitation of CP accumulation is missing from this figure. In the case of other VSR4 mutants as well as when over expressed or down regulated, the authors demonstrate viral RNA accumulation. How does OE of VSR-C1A mutant alter the (+)- and (-)-strand synthesis? I don’t agree yet with the statement that VSR4-6K2 interaction is critical for TuMV infection.

**Part III – Minor Issues: Editorial and Data Presentation Modifications**

Reviewer #1: (No Response)

Reviewer #2: Chloroplast fluorescence is hardly visible in Fig 3C 48h. It is difficult to detect the signal also in 3E Col FM4-64 and avr4 mutant mCherry ARA Q69I panels.

PLOS authors have the option to publish the peer review history of their article (what does this mean?). If published, this will include your full peer review and any attached files.

Reviewer #1: No

Reviewer #2: No

Figure Files:

Data Requirements:

Reproducibility:

References:

---

## [Editor Report · Decision Letter 1]

7 Jan 2022

Dear Dr. Yan,

We are pleased to inform you that your manuscript 'Turnip mosaic virus co-opts the vacuolar sorting receptor VSR4 to promote viral genome replication in plants by targeting viral replication vesicles to the endosome' has been provisionally accepted for publication in PLOS Pathogens.

Best regards,

Savithramma P. Dinesh-Kumar

Associate Editor

PLOS Pathogens

Peter Nagy

Section Editor

PLOS Pathogens

Kasturi Haldar

Editor-in-Chief

PLOS Pathogens

orcid.org/0000-0001-5065-158X

Michael Malim

Editor-in-Chief

PLOS Pathogens

orcid.org/0000-0002-7699-2064
---

## [Editor Report · Acceptance letter]

20 Jan 2022

Dear Dr. Yan,

We are delighted to inform you that your manuscript, "Turnip mosaic virus co-opts the vacuolar sorting receptor VSR4 to promote viral genome replication in plants by targeting viral replication vesicles to the endosome," has been formally accepted for publication in PLOS Pathogens.

Best regards,

Kasturi Haldar

Editor-in-Chief

PLOS Pathogens

orcid.org/0000-0001-5065-158X

Michael Malim

Editor-in-Chief

PLOS Pathogens

orcid.org/0000-0002-7699-2064